# shRNA-mediated down-regulation of Acsl1 reverses skeletal muscle insulin resistance in obese C57BL6/J mice

Kamila Roszczyc-Owsiejczuk[1], Monika Imierska[1], Emilia Sokołowska[1], Mariusz Kuźmicki[2], Karolina Pogodzińska[1], Agnieszka Błachnio-Zabielska[1☯*], Piotr Zabielski[3☯*]

**1** Department of Hygiene, Epidemiology and Metabolic Disorders, Medical University of Bialystok, Bialystok, Poland, **2** Department of Gynecology and Gynecological Oncology, Medical University of Bialystok, Bialystok, Poland, **3** Department of Medical Biology, Medical University of Bialystok, Bialystok, Poland

☯ These authors contributed equally to this work.
* piotr.zabielski@umb.edu.pl (PZ); agnieszka.blachnio-zabielska@umb.edu.pl (AB-Z)

## Abstract

Prolonged consumption of diet rich in fats is regarded as the major factor leading to the insulin resistance (IR) and type 2 diabetes (T2D). Emerging evidence link excessive accumulation of bioactive lipids such as diacylglycerol (DAG) and ceramide (Cer), with impairment of insulin signaling in skeletal muscle. Until recently, little has been known about the involvement of long-chain acyl-CoAs synthetases in the above mechanism. To examine possible role of long-chain acyl-coenzyme A synthetase 1 (Acsl1) (a major muscular ACSL isoform) in mediating HFD-induced IR we locally silenced Acsl1 in gastrocnemius of high-fat diet (HFD)-fed C57BL/6J mice through electroporation-delivered shRNA and compared it to non-silenced tissue within the same animal. Acsl1 down-regulation decreased the content of muscular long-chain acyl-CoA (LCACoA) and both the Cer (C18:1-Cer and C24:1-Cer) and DAG (C16:0/18:0-DAG, C16:0/18:2-DAG, C18:0/18:0-DAG) and simultaneously improved insulin sensitivity and glucose uptake as compared with non-silenced tissue. Acsl1 down-regulation decreased expression of mitochondrial β-oxidation enzymes, and the content of both the short-chain acylcarnitine (SCA-Car) and short-chain acyl-CoA (SCA-CoA) in muscle, pointing towards reduction of mitochondrial FA oxidation. The results indicate, that beneficial effects of Acsl1 partial ablation on muscular insulin sensitivity are connected with inhibition of Cer and DAG accumulation, and outweigh detrimental impact of decreased mitochondrial fatty acids metabolism in skeletal muscle of obese HFD-fed mice.

## Introduction

Excessive caloric intake in a form of diet rich in fats (HFD) is associated with accumulation of intramuscular lipids (IMCLs), leading to insulin resistance (IR) and type 2 diabetes (T2D) [1]. Prior to entering molecular pathways, diet-derived plasma free fatty acids (FFA) are activated intracellularly to long-chain acyl-CoA (LCACoA) by one of the members of the family of

**Data Availability Statement:** All relevant data are within the manuscript and its Supporting Information files.

**Funding:** This research was funded by a Foundation for Polish Science, grant number TEAM/2016-1/2 and Medical University of Bialystok, grant numbers SUB/1/DN/20/003/1117 and SUB/1/DN/21/002/1204. The funders had no role in study design, data collection and analysis, decision to publish, or preparation of the manuscript.

**Competing interests:** The authors have declared that no competing interests exist.

long-chain acyl-CoA synthetases (ACSLs). Five ACSL isoenzymes have been found in mammalian tissues [2], with distinct tissue distribution, subcellular localization and substrate specificity [3]. In skeletal muscle ACSL1 is responsible for approx. 90% of total LCACoA synthesis [4]. Its subcellular localization is distributed between mitochondria and smooth endoplasmic reticulum, which suggests that ACSL1 lays at the crossroads between lipid synthesis and mitochondrial lipid β-oxidation as a control point in the muscular LCACoAs metabolism [5].

It has been evidenced that LCACoA modulate skeletal muscle insulin sensitivity through various mechanisms. Excess of LCACoA induces accumulation of oxidation byproducts such as short-chain acyl carnitines (SCA-Car) and short-chain acyl-CoAs (SCACoA) which negatively affects mitochondrial metabolism and insulin action [6]. On the other hand, LCACoA modulate it indirectly, via its involvement in de-novo synthesis of diacylglycerol (DAG) or ceramide (Cer). DAG leads to activation of PKC isoforms that promote serine/threonine phosphorylation of insulin receptor substrate 1 (IRS-1). Concomitant accumulation of intramuscular LCACoA and DAG was shown in skeletal muscles of obese and insulin-resistant Zucker fa/fa rats [7] and in insulin-resistant rats fed high-fat diet [8]. Ceramide can hinder insulin signaling in two ways: first, by disrupting the insulin signaling pathway through the protein phosphatase 2A (PP2A), leading to reduced phosphorylation and activity of protein kinase B (Akt/PKB); second, by preventing the movement of serine/threonine kinase Akt/PKB to the cell membrane, achieved through a mechanism involving atypical protein kinase Cζ (PKCζ). Both mechanisms are presently well investigated and considered accountable for the immediate impacts of lipid oversupply on tissue insulin sensitivity [9–13].Intramuscular content of Cer was shown to increase in the muscles of IR Zucker rats [7], Zucker diabetic fatty (ZDF) rats [14] and obese humans [15]. Recent studies suggest that C18:0-Cer and C18:1-Cer species are the most likely culprit regarding direct inhibition of skeletal insulin signaling pathway [16–18]. ACSL1 displays high substrate affinity towards major plasma-derived fatty acids such as palmitate (C16:0 FA), stearate (C18:0 FA) and oleate (C18:1 FA) [19]. This makes Acsl1 especially crucial in the synthesis of both the sphingosine (from C16:0-CoA and L-Serine) and C18:0-Cer and C18:1-Cer (from sphingosine and C18:0- and C18:1-CoAs). Despite interdependence of acyl-CoA metabolism and signaling lipids, the exact connection between LCACoA metabolic flux, fatty acids β-oxidation, accumulation of IMCLs and IR is still under debate.

The objective of this study was to determine the role of ACSL1-derived LCACoA in the induction of skeletal muscle IR through modulation of muscular bioactive lipids such as DAG and Cer in C57BL6/J mice. The major factor influencing our choice of C57BL/6J mouse strain was both the well documented susceptibility to metabolic disorders under HFD feeding and its widely recognized status in insulin resistance and T2D research [20,21]. C57BL6/J strain's predisposition to develop insulin resistance makes it a preferred model for studying metabolic disorders induced by high-fat diets [22]. This metabolic feature may partially arise from C57BL/6J-specific mutation in Nnt gene leading to the expression of non-functional mitochondrial nicotinamide nucleotide transhydrogenase [23,24]. Although the most significant phenotypic variation between C57BL6/J and other BL6 strains is β-cell energy metabolism and insulin release, the effect of Nnt deletion seem to be moderate in nature [25,26]. Control C57BL6/J mice lacking functional Nnt gene housed on standard, low-fat diet are still significantly more glucose-tolerant and insulin sensitive from their high-fat diet-fed counterparts, which validates the use of C57BL/6J strain in the studies on high-fat diet-induced insulin resistance [23,25]. Consequently, the utilization of the C57BL/6J mice, with its established predisposition to glucose intolerance and reduced insulin secretion under high-caloric intake, offers a robust model for investigating metabolic disorders.

We employed a model of in-vivo shRNA-mediated, electroporation delivered gene silencing which is often used in the studies of muscle function [27–29]. Within the same HFD-fed, insulin-resistant C57BL/6J mouse, one gastrocnemius received scrambled shRNA construct, yielding tissue with intact Acsl1 expression (HFD$_{(+Acsl1)}$ muscle), whereas gastrocnemius from the contralateral hindlimb was electroporated with active shRNA plasmid, yielding Acsl1-down-regulated muscle (HFD$_{(-Acsl1)}$ muscle). This approach differs significantly from the experiments employing whole-body or muscle-specific Acsl1 knock-outs, as local silencing in single muscle does not affect whole-body metabolism. The similar experimental model was employed by other research groups in the studies on muscle function and metabolism [30–33]. We also collected gastrocnemius muscle form low-fat diet (LFD) counterparts with intact Acsl1 expression (LFD$_{(+Acsl1)}$ muscle) as insulin-sensitive reference.

To study the effect of Acsl1 silencing in skeletal muscle of HFD-fed mice, we studied FA uptake and mitochondrial FA channeling, we measured protein expression of fatty acids transporters, CPT1B expression and the content of acyl-CoAs and acyl-carnitines. We established Cer and DAG content to determine the effects of Acsl1 modulation on the muscular bioactive lipid accumulation. Finally, we determined activation state of insulin phosphorylation cascade and muscular glucose uptake in both tissue types. Our results indicate, that Acsl1 down-regulation in insulin-resistant muscle significantly decreases accumulation of both the DAG and Cer and augments muscular insulin signaling and glucose uptake, pointing to Acsl1-derived LCACoAs as the possible optimal target candidate for the treatment of muscular IR.

## Materials and methods

### Animals

The research was conducted on male C57BL/6J mice purchased from Jackson Laboratory (Bar Harbor, ME, USA) of 6 weeks of age at the beginning of the experiment. The mice were housed in individually ventilated cages (IVCs) at 21˚C using a 12 h light /12 h dark cycle. After an adaptation period, mice were randomly divided into the low-fat diet group, fed a standard rodent diet (LFD, Research Diets INC D12450J, n = 8) and the high-fat diet group (HFD, n = 8, Research Diets INC, D12492). The mice had ad-libitum access to food and water. All animals received appropriate diet for 8 weeks. All animal procedures were performed according to approval 35/2016 issued by the Local Ethical Committee for Animal Experiments, Olsztyn, Poland. The reporting in the manuscript and the supporting information follows the recommendations in the ARRIVE guidelines.

### In vivo muscular Acsl1 down-regulation

The gene silencing procedure was performed during the 2$^{nd}$ week of the experiment. Gastrocnemius intramuscular injection of plasmid DNA followed by electroporation was performed under isoflurane anesthesia, as described previously [28]. Animals received lidocaine ointment at the site of electrode placement and ibuprofen analgesia in drinking water (0.5mg/ml, for 24 hours) to decrease post-electroporation muscle sore. Plasmids contained Turbo green fluorescent reporter protein (TurboGFP) and appropriate silencing or scrambled shRNA sequences controlled by murine mCMV promoter. Plasmid stocks were produced in recombinant E. coli cultures (acquired from Dharmacon, currently Horizon Discovery, Cambridge, UK). After an overnight growth of the bacterial culture, high-quality plasmid DNA was isolated in accordance with the GeneJET Plasmid Maxiprep Kit protocol (Thermo Scientific, Waltham, MA, USA). Before in vivo electroporation, a mix of three different target shRNA sequences at a total concentration of 2 μg/μL in 150 mM PBS (pH = 7.2) was prepared.

To study the effects of Acsl1 down-regulation in HFD-fed animals, left gastrocnemius muscle of insulin resistant high-fat diet mice was electroporated with active shRNA plasmid (#V3SM11241-230929842, #V3SM11241-232909593, #V3SM11241-237032778, Dharmacon, currently Horizon Discovery, Cambridge, UK) yielding down-regulated HFD$_{(-Acsl1)}$ gastrocnemius. The contralateral, right hindlimb gastrocnemius within the same animal was transfected with scrambled shRNA (#VSC11708), yielding HFD$_{(+Acsl1)}$ gastrocnemius, used in HFD$_{(+Acsl1)}$ vs HFD$_{(-Acsl1)}$ comparison. To observe the effect of HFD-feeding on skeletal muscle insulin sensitivity, mice fed low-fat diet received scrambled shRNA plasmid (#VSC11708), yielding LFD$_{(+Acsl1)}$ tissue with intact Acsl1 expression, used in LFD$_{(+Acsl1)}$ vs HFD$_{(+Acsl1)}$ comparison. The experimental study design is presented in **S1 Fig**.

The expression of the GFP reporter gene in electroporated muscle was monitored transcutaneously every week with a UV flashlight. **S2 Fig** presents exemplary visualization of TurboGFP reporter gene expression in muscle at the end of experiment. There was no difference in gastrocnemius weigh between silenced and non-silenced muscle. Cross-sectional slides had shown normal muscle morphology, except minute necrosis around needle marks visible in both the silenced and non-silenced muscle.

At the end of the experiment (**S1 Fig**), the animals were euthanized by cervical dislocation, according to local IACUC approval. Directly after the euthanasia the gastrocnemius muscles were taken, immediately frozen in liquid nitrogen and stored at -80˚C until assayed.

## Oral glucose tolerance test (OGTT) and Insulin tolerance test (ITT), Plasma glucose, insulin, HOMA-IR

Two weeks before sacrifice, OGTT was performed in conscious mice after 6 hours fasting. Animals received oral glucose gavage at a dose of 2 g/kg. Blood glucose concentration were measured from the tail vein blood using Accu-Chek Aviva glucometer (Roche). ITTs were performed in similar conditions 1 week after OGTT, with intraperitoneal insulin injection (NovoRapid) at a dose of 0.75 U/kg body weight. The area under the glucose concentration curve for both the OGTT and ITT was measured according to the trapezoidal rule. Plasma insulin concentrations were assayed using the ELISA insulin assay (Rat/Mouse Insulin Millipore, Merck KGaA) according to the manufacturer's instructions. Homeostatic Model Assessment of Insulin Resistance (HOMA-IR) was calculated according to the formula provided by Cacho et al. [34]: HOMA-IR = [fasting glucose (mg/dL) x fasting insulin (µIU/mL)]/2430. **S3 Fig** presents the results of the above measurements.

## Insulin-stimulated glucose uptake

Glucose uptake was estimated with intravenous bolus of 2-deoxy-[1,2-3H (N)]-D-glucose according to the previous published protocol [35]. Detailed description of measurements and calculations are provided in the online **S1 Appendix** **Materials and Methods** (section Insulin-stimulated glucose uptake). Exemplary plasma glucose concentration, 2-deoxy-[1,2-3H (N)]-D-glucose radioactivity and plasma tracer enrichment curves after bolus injection of the tracer under insulin stimulation are presented in **S4 Fig**.

## Lipid measurements

Plasma FFA was measured according to the method described by Persson et al. [36], using UHPLC/MS/MS. LCACoA as well as malonyl-CoA and other SCACoAs were extracted according to Minkler et al. [37]. Acyl-CoAs were measured according to Blachnio-Zabielska et al. [38]. Both the SCA-Car and long-chain acyl-carnitines (LCA-Car) were assayed according to Gies-bertz et al. [39] with minor modifications, using the UHPLC/MS/MS method with C17:0-acyl carnitine as internal standard. DAG and Cer tissue content was established by

UHPLC/MS/MS methods by Blachnio-Zabielska et al. [40,41]. Details of measurements are provided in the online **S1 Appendix Materials and Methods**. Intramuscular and plasma content of TAG was measured with a High Sensitivity Triglyceride Fluorometric Assay Kit (MAK264-1KT, Merck KGaA).

## Western blot

Western-blot was used to assess protein expression of the muscular FA transporters, CPT1B, mitochondrial β-oxidation proteins, acetyl-CoA carboxylase (ACC) phosphorylation, efficiency of Acsl1 shRNA silencing at the protein level, glucotransporter 4 (GLUT4) and the activation state of insulin signaling cascade. Details of measurements and antibodies used in the study are provided in the online **S1 Appendix Materials and Methods** and **S1 Table**, respectively.

## Real-time PCR (RT-PCR)

To estimate the efficiency of shRNA silencing of Acsl1 gene, and the impact of Acsl1 silencing on the expression of other muscular isoforms of ACSL (2 to 6) we performed RT-PCR measurements. Total RNA was isolated from pulverized samples using a mirVana Isolation Kit (Thermo-Scientific) according to the manufacturer's guidelines. Reverse transcription was performed with the use of a Transcriptor First Strand cDNA Synthesis Kit (Roche). Acsl1 and GAPDH gene expression levels were analyzed via a real-time polymerase chain reaction (RT-PCR) with the use of RealTime ready Custom Assays and a LightCycler480 system (Roche). The results were normalized to housekeeping gene GAPDH expression. Sequence of primers is given in **S2 Table**.

## Statistical analysis

Statistical significance was established using non-parametric Wilcoxon signed rank test for $HFD_{(+Acsl1)}$ and $HFD_{(-Acsl1)}$ comparison (paired samples, within-animal, non-silenced and silenced contralateral muscles, respectively). Non-parametric Wilcoxon rank sum test was used for $LFD_{(+Acsl1)}$ and $HFD_{(+Acsl1)}$ comparison (non-paired samples, non-silenced muscle, from low-fat and high-fat diet mice, respectively). Values are presented as medians and interquartile ranges. Statistical significance was determined by $p < 0.05$. Asterisks are used to indicate statistical significance; ns refers to $p > 0.05$; $* \leq 0.05$; $** \leq 0.01$. The calculations were computed using GraphPad Prism 8.0.2.

## Results

### HFD diet robustly induces obesity and systemic insulin resistance in C57BL/6J mice

HFD-fed animals displayed significant increase in body weight and systemic insulin resistance as evidenced by impaired glucose tolerance, reduced insulin responsiveness and an increase in the HOMA-IR index value, as compared to the LFD group (**S3 Fig** and **S3 Table**). Moreover, HFD-fed mice displayed lower WBC and significantly higher platelet counts in blood as compared to LFD animals (**S3 Table**), which points toward obesity-induced prothrombotic state. Both, total plasma TAG and FFA concentration significantly increased in the HFD-fed animals compared to the LFD counterparts (**S3 Table**).

### shRNA electroporation down-regulates ACSL1 at mRNA and protein level in gastrocnemius of HFD-fed mice

$HFD_{(+Acls1)}$ gastrocnemius displayed increased content ACSL1 at the level of mRNA and protein in the as compared to the $LFD_{(+Acsl1)}$ muscle (**Fig 1, Panel A1—B1**). In HFD-fed animals,

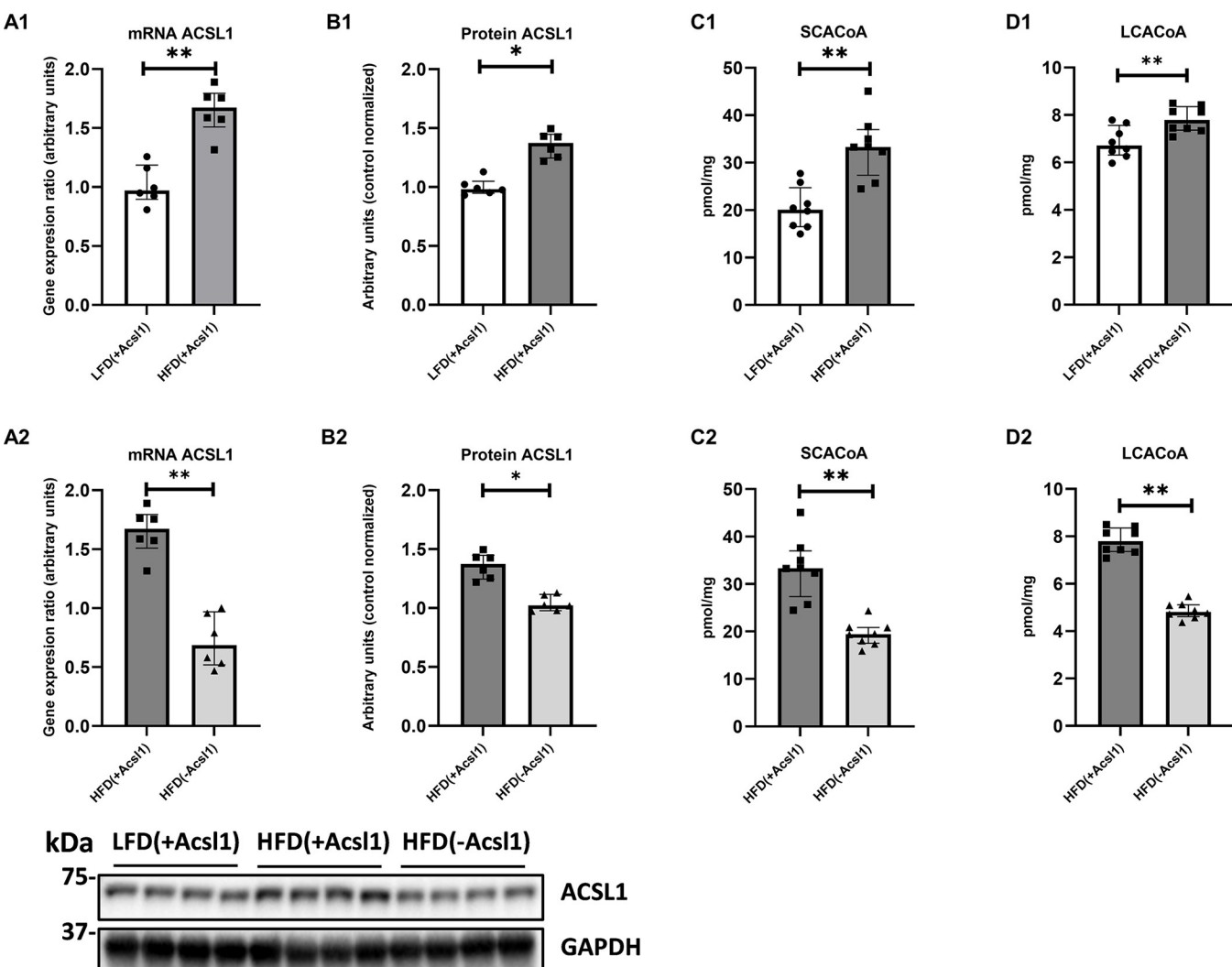

**Fig 1. The effect of skeletal muscle Acsl1 partial ablation on the protein and gene expression of ACSL1 and muscular concentration of acyl-CoAs.** Panel (**A1-A2**)—mRNA expression of ACSL1; Panel (**B1-B2**)—protein expression of Acsl1; Panel (**C1-C2**) and (**D1-D2**)—the content of SCA-CoA and LCACoA, respectively. LFD(+Acsl1)—gastrocnemius from LFD-fed mice, with intact Acsl1 expression (scrambled plasmid); HFD(+Acsl1)—gastrocnemius from HFD-fed mice with intact Acsl1 expression (scrambled plasmid); HFD(-Acsl1)—contralateral hindlimb gastrocnemius from HFD-fed mice, with down-regulated Acsl1 expression (silencing shRNA plasmid). Panels A1 to D1 present effect of diet (LFD(+Acsl1) and HFD(+Acsl1) muscle). Panels A2 to D2 present effect of Acsl1 silencing within HFD-fed animals (HFD(+Acsl1) vs HFD(-Acsl1) muscle). Values are median ± interquartile range; n = 6 per group (mRNA and protein); n = 8 per group (other data). * -p ≤ 0.05; **- p ≤ 0.01.

Acsl1 gene silencing significantly decreased both, the mRNA (by approx. 61%) and protein content (by approx. 23%) of ACSL1 as compared to the contralateral, non-silenced HFD(+Acsl1) muscle (p < 0.05) (**Fig 1, Panel A2—B2**). We did not observe significant compensatory up-regulation in the expression of other muscular ACSL isoforms between silenced and non-silenced muscle of HFD-fed animals (**S5 Fig**).

## Down regulation of muscular ACSL1 decreases HFD-induced accumulation muscular SCACoA and LCACoA

The total content of both, SCACoA and LCACoA significantly increased in HFD(+Acsl1) muscle compared to LFD(+Acsl1) tissue (**Fig 1, Panel C1—D1; S4 Table**).

In HFD-fed animals Acsl1 silencing in HFD$_{(-Acsl1)}$ gastrocnemius significantly decreased total content of both SCACoA and LCACoA as compared to contralateral HFD$_{(+Acsl1)}$ muscle ($p < 0.05$). The strongest reduction of individual LCACoA species between HFD$_{(+Acsl1)}$ and HFD$_{(-Acsl1)}$ gastrocnemius was observed for C16:0-CoA, C18:2-CoA and C24:0-CoA ($p < 0.05$ for all cases) (**Fig 1, Panel C2—D2; S5 Table**).

## ACSL1 modulates expression of muscular FATP1 and FABPpm fatty acid transporters in muscle from HFD-fed animals

The content of all studied FA transporters significantly increased in HFD$_{(+Acsl1)}$ gastrocnemius compared to LFD$_{(+Acsl1)}$ counterpart ($p < 0.05$), (**Fig 2, Panel, A1-C1**). Acsl1down-regulation in within HFD-fed animals had no significant effect on CD36 expression, yet it significantly elevated FABPpm protein above HFD$_{(+Acsl1)}$ values ($p < 0.05$). Silencing of Acsl1 down-regulated FATP1 protein in HFD$_{(-Acsl1)}$ muscle as compared to contralateral HFD(+Acsl1) tissue ($p < 0.05$) (**Fig 2, Panel A2-C2**).

## ACSL1 down-regulation modulates the expression of enzymes of mitochondrial lipid metabolism in muscle from obese animals

The protein content of CPT1B was significantly decreased in HFD$_{(+Acsl1)}$ muscle compared to LFD$_{(+Acsl1)}$ tissue ($p < 0.05$). It was accompanied by increased inhibition of ACC (through its phosphorylation) (**Fig 2, Panel D1, E1**). Concomitantly, the expression of mitochondrial β-oxidation enzymes eg. trifunctional enzyme (HADHA), long-chain (ACADVL) and medium-chain (ACADM) acyl-CoA dehydrogenases was significantly up-regulated in HFD$_{(+Acsl1)}$ muscle compared to LFD$_{(+Acsl1)}$ tissue ($p < 0.05$) (**Fig 2, Panel F1-H1**).

Acsl1 silencing in muscle of HFD-fed animals up-regulated CPT1B expression, and simultaneously decreased ACC phosphorylation ($p < 0.05$) (**Fig 2, Panel D2, E2)** and protein expression of the mitochondrial β-oxidation enzymes (**Fig 2, Panel F2-H2).**

## ACSL1 silencing decreases muscular content of SCA-Car and LCA-Car in muscle from obese mice

HFD feeding modestly increased the content of LCA-Car and decreased SCA-Car in HFD$_{(+Acsl1)}$ muscle compared to LFD$^{(+Acsl1)}$ values (**Fig 2, Panel I1-J1, S4 Table**). Within HFD-fed animals Acsl1 down-regulation significantly decreased both the SCA-Car and LCA-Car by approx. 50% as compared to contralateral HFD$_{(+Acsl1)}$ muscle ($p < 0.05$). The greatest reduction LCA-Car species was observed for C14:0-Car, C16:0-Car, C18:0-Car and C18:1-Car ($p < 0.05$ for all cases) (**Fig 2, Panel I2-J2; S5 Table**).

## ACSL1 down-regulation in skeletal muscle of HFD-fed mice alleviates accumulation of TG, DAG and Cer

The content of TAG, DAG and Cer in the HFD$_{(+Acsl1)}$ gastrocnemius was significantly higher compared to LFD$_{(+Acsl1)}$ muscle ($p < 0.01$ in all cases) (**Fig 3, Panels A1-C1**). Regarding DAG and Cer molecular species, highest increase was noted for C16:0/18:0-DAG, C16:0/18:2-DAG, C18:0/18:2-DAG, 18:0/20:0-DAG and C16:0-Cer, C18:0-Cer, C18:1-Cer, C20:0-Cer, C22:0-Cer, C24:1-Cer and C24:0-Cer, respectively (**S4 Table**).

Acsl1 silencing in HFD$_{(-Acsl1)}$ gastrocnemius decreased total content of TAG, DAG and Cer below the values observed in non-silenced contralateral HFD$_{(+Acsl1)}$ muscle ($p<0.01$ in all cases) (**Fig 3, Panel A2-C2**). The highest decrease was observed for C16:0/18:0, C16:0/18:1,

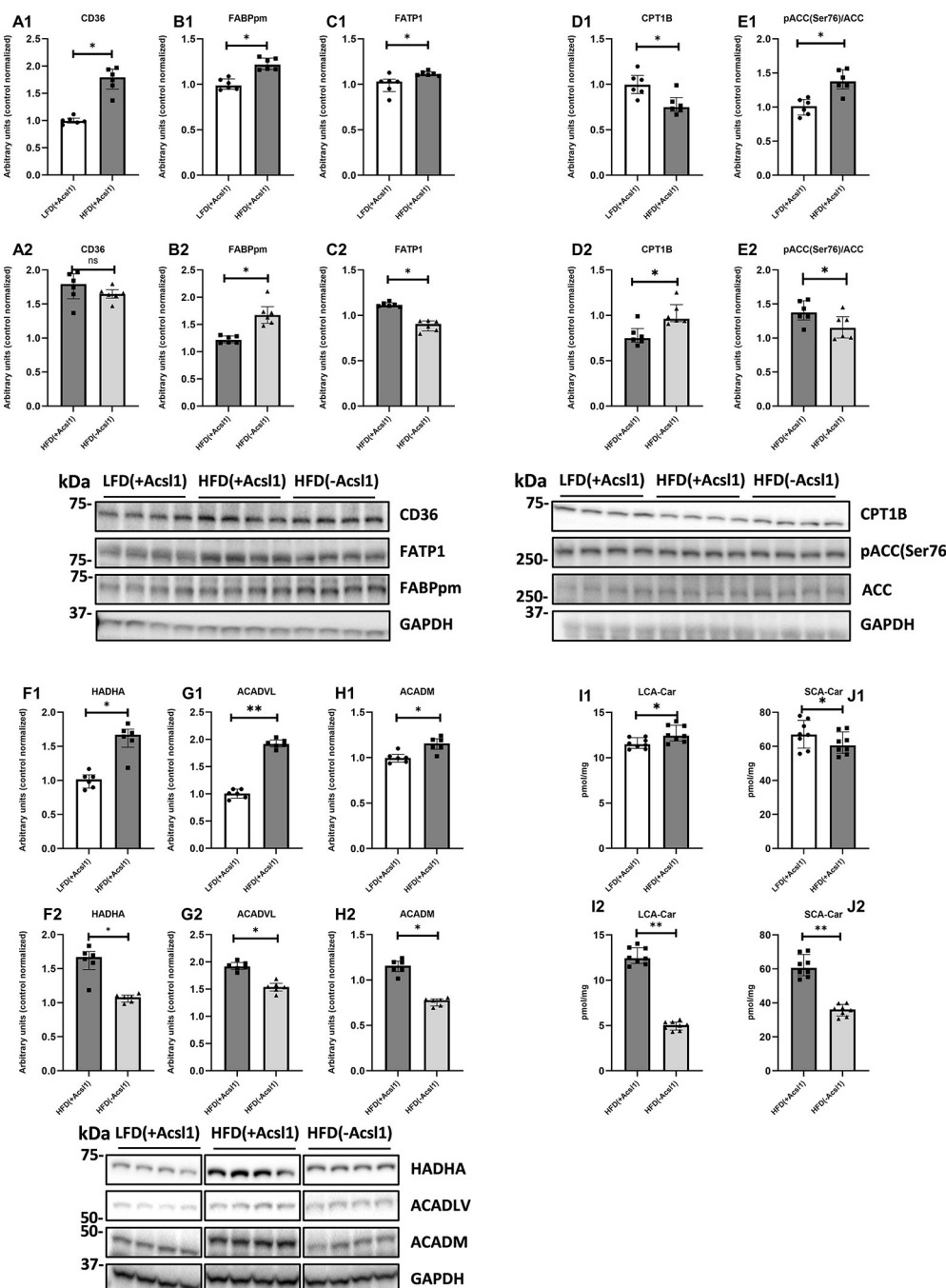

**Fig 2. The effect of skeletal muscle Acsl1 down-regulation on the expression of fatty acid transporters, CPT1B, mitochondrial β-oxidation proteins, ACC phosphorylation and the content of acyl-carnitines in the gastrocnemius of high-fat-diet-fed mice.** Panels (**A1-C1; A2-C2**)—protein expression of CD36, FABPpm, FATP; Panel (**D1-D2**)—protein expression of CPT1B; Panel (**E1-E2**)—phosphorylation state of ACC protein; Panels (**F1-H1; F2-H2**)–protein expression of mitochondrial β-oxidation trifunctional enzyme, very long- and medium-chain acyl-CoA dehydrogenases (HADHA, ACADVL and ACADM, respectively); Panels (**I1-J1; I2-J2**)—the content of short-chain (SCA-Car) and long-chain (LCA-Car) acyl-carnitines. LFD$_{(+Acsl1)}$–gastrocnemius from LFD-fed mice, with intact Acsl1 expression (scrambled plasmid); HFD$_{(+Acsl1)}$—gastrocnemius from HFD-fed mice with intact Acsl1 expression (scrambled plasmid); HFD(-Acsl1)–contralateral hindlimb gastrocnemius from HFD-fed mice, with down-regulated Acsl1 expression (silencing shRNA plasmid). Panels A1 to J1 present effect of diet (LFD$_{(+Acsl1)}$ and HFD$_{(+Acsl1)}$ muscle). Panels A2 to J2 present effect of Acsl1 silencing within HFD-fed animals (HFD$_{(+Acsl1)}$ vs HFD$_{(-Acsl1)}$ muscle).Values are median ± interquartile range; n = 6 per group (protein); n = 8 per group (other data). ns -p > 0.05; * -p ≤ 0.05; **- p ≤ 0.01.

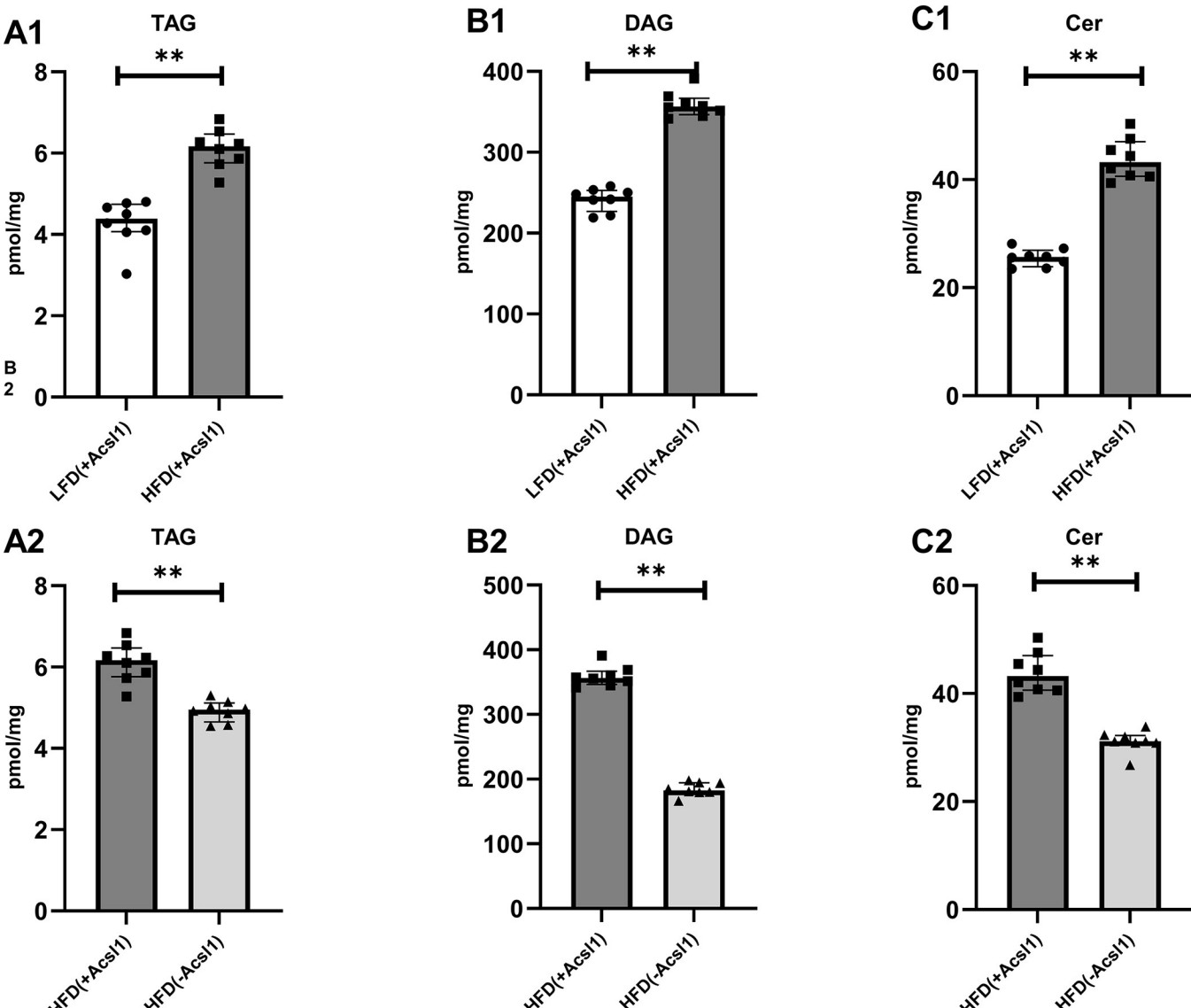

**Fig 3. Total lipid content in gastrocnemius in Acsl1-silenced mouse gastrocnemius.** Panel (**A1-A2**)—total content of triacylglycerols (TAG); Panel (**B1-B2**) —total content of diacylglycerols (DAG); Panel (**C1-C2**)—total content of ceramide (Cer). LFD$_{(+Acsl1)}$−gastrocnemius from LFD-fed mice, with intact Acsl1 expression (scrambled plasmid); HFD$_{(+Acsl1)}$—gastrocnemius from HFD-fed mice with intact Acsl1 expression (scrambled plasmid); HFD(-Acsl1) −contralateral hindlimb gastrocnemius from HFD-fed mice, with down-regulated Acsl1 expression (silencing shRNA plasmid). Panels A1 to C1 present the effect of diet (LFD$_{(+Acsl1)}$ vs HFD$_{(+Acsl1)}$ muscle). Panels A2 to C2 present effect of Acsl1 silencing within HFD-fed animals (HFD$_{(+Acsl1)}$ vs HFD$_{(-Acsl1)}$ muscle). Values are median ± interquartile range; n = 8 per group. **- p ≤ 0.01.

C16:0/18:2, C18:0/18:0, C18:0/18:1-DAG, C18:1/18:1-DAG and C18:2/18:2 DAG species and C18:1-Cer and C24:1-Cer (**S5 Table**).

## Down-regulation of ACSL1 improves insulin signaling and insulin-stimulated glucose uptake in skeletal muscle from obese, HFD-fed mice

High-fat diet significantly impaired insulin-stimulated signaling in HFD$_{(+Acsl1)}$ gastrocnemius from HFD-fed mice compared to the LFD$_{(+Acsl1)}$ gastrocnemius from LFD-fed animals. It was evidenced by decreased insulin receptor and IRS-1 activatory phosphorylation (**Fig 4, Panel**

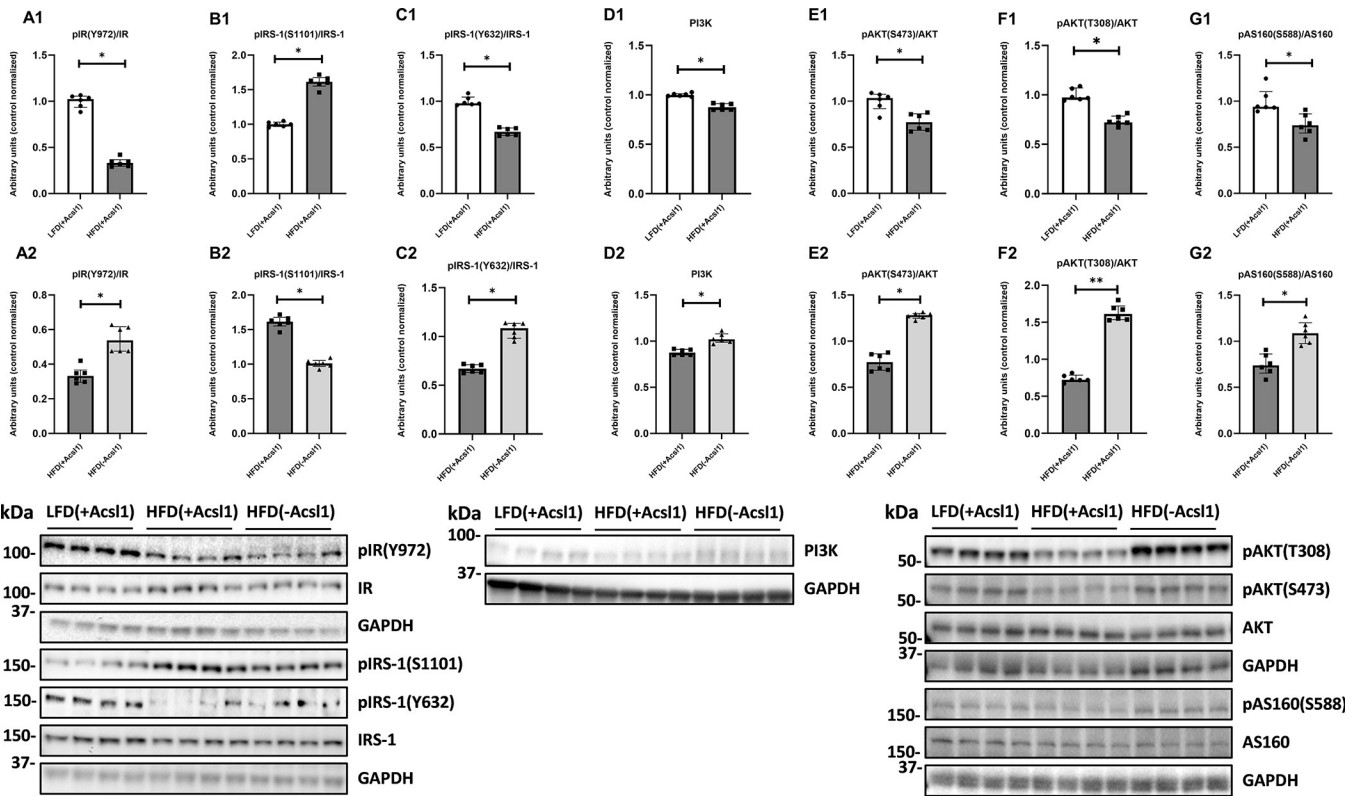

**Fig 4. Insulin signaling pathway in Acsl1-silenced mouse gastrocnemius.** Panel (**A1-A2**)—insulin receptor phosphorylation (pIR Y972); Panel (**B1-B2**)—serine phosphorylation (pIRS-1 S1101) and Panel (**C1-C2**) tyrosine phosphorylation (pIRS-1 Y632) of insulin receptor substrate 1 (IRS-1); Panel (**D1-D2**)—protein expression of phosphoinositide 3-kinase (PI3K); Panel (**E1-E2**)—serine phosphorylation of Akt/ protein kinase B (pAKT S473); Panel (**F1-F2**)—threonine phosphorylation of Akt/ protein kinase B (pAKT T308); Panel (**G1-G2**)—serine phosphorylation of Akt/PKB 160kDa substrate (pAS160 S588). LFD$_{(+Acsl1)}$−gastrocnemius from LFD-fed mice, with intact Acsl1 expression (scrambled plasmid); HFD$_{(+Acsl1)}$—gastrocnemius from HFD-fed mice with intact Acsl1 expression (scrambled plasmid); HFD(-Acsl1)−contralateral hindlimb gastrocnemius from HFD-fed mice, with down-regulated Acsl1 expression (silencing shRNA plasmid). Panels A1 to F1 present the effect of diet (LFD$_{(+Acsl1)}$ vs HFD$_{(+Acsl1)}$ muscle). Panels A2 to F2 present effect of Acsl1 silencing within HFD-fed animals (HFD$_{(+Acsl1)}$ vs HFD$_{(-Acsl1)}$ muscle). Values are median ± interquartile range; n = 6 per group. * -p $\leq$ 0.05.

A1, C1; p<0.05), increased in inhibitory phosphorylation of S1101of IRS (**Fig 4**, **Panel B1;** p<0.05), down-regulation of PI3K (**Fig 4** **Panel D1,** p<0.05) and decreased phosphorylation of AKT (at both the serine S473 and threonine T308 sites) and its substrate AS160 (**Fig 4, Panels E1 and F1;** p<0.05). In line with above findings we observed significant down regulation of GLUT4 expression and inhibition of insulin-stimulated glucose uptake in HFD$_{(+Acsl1)}$ gastrocnemius compared to LFD(+Acsl1) tissue (**Fig 5, Panels A1 and B1; p<0.05**).

ACSL1 down-regulation in HFD$_{(-Acsl1)}$ muscle from HFD-fed mice improved the insulin signaling at all of the studied levels (**Fig 4, Panels A2 to E2**) and both the GLUT4 expression and insulin-stimulated glucose uptake (**Fig 5, Panels A2 and B2**) as compared to non-silenced contralateral HFD$_{(+Acsl1)}$ gastrocnemius.

## Discussion

Compared to other insulin-sensitive tissues, ACSL physiology in skeletal muscle under lipid overload was not studied extensively. Skeletal muscle shows the expression of several ACSL isoforms [42] of which isoform 5 and 6 was studied previously with the use of plasmid-based overexpression [43] or siRNA-mediated silencing [44] in cell-based, in-vitro experiments. As Acsl5 and 6 are expressed in significantly lower quantities than Acsl1, and the latter is

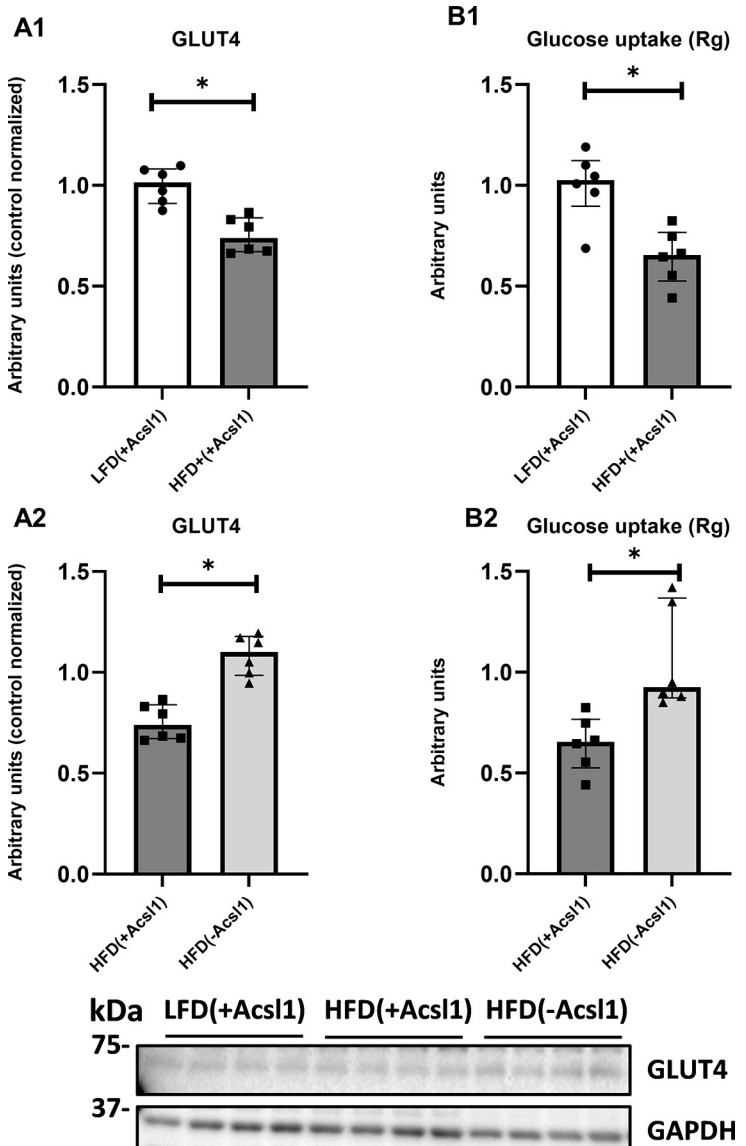

**Fig 5. The effect of Acsl1 partial ablation on mouse gastrocnemius muscle insulin-stimulated glucose uptake.**
Panel (**A1-A2**)—protein expression of glucotransporter 4 (GLUT4); Panel (**B1-B2**)—insulin-stimulated glucose uptake. LFD$_{(+Acsl1)}$—gastrocnemius from LFD-fed mice, with intact Acsl1 expression (scrambled plasmid); HFD$_{(+Acsl1)}$—gastrocnemius from HFD-fed mice with intact Acsl1 expression (scrambled plasmid); HFD(-Acsl1)—contralateral hindlimb gastrocnemius from HFD-fed mice, with down-regulated Acsl1 expression (silencing shRNA plasmid). Panels A1 and B1 present the effect of diet (LFD$_{(+Acsl1)}$ vs HFD$_{(+Acsl1)}$ muscle). Panels A2 and B2 present effect of Acsl1 silencing within HFD-fed animals (HFD$_{(+Acsl1)}$ vs HFD$_{(-Acsl1)}$ muscle). Values are median ± interquartile range; n = 6 per group. * -p ≤ 0.05.

responsible for 90% of total ACSL activity, we selected Acsl1 isoform as the most suitable target for bioactive lipid modulation in skeletal muscle. Previous studies had shown, that muscle-specific knock-out of ACSL1 decreases mitochondrial FA oxidation [4], yet improves overall whole-body insulin sensitivity [45]. It has to be noted that none of the previous studies were performed under lipid-overload and IR condition and did not measure bioactive lipid content. As induction of skeletal muscle insulin resistance is correlated with excessive fat accumulation and obesity, it is important to assess the ACSL1-mediated lipid metabolism under the HFD-

induced IR state. To address this question, we locally down- regulated ACSL1 in mouse gastrocnemius from obese, insulin resistant HFD-fed C57BL6/J mice. In our model, one hindlimb gastrocnemius was electroporated with active shRNA silencing plasmid (HFD$_{(-Acsl1)}$ muscle), whereas the contralateral hindlimb received scrambled shRNA plasmid (HFD$_{(+Acsl1)}$ muscle). Similar experimental model was employed also by other research groups to study various aspects of muscle function and metabolism [30–33].

Our study show that local ACSL1 down-regulation in HFD$_{(-Acsl1)}$ muscle improves insulin sensitivity compared to HFD$_{(+Acsl1)}$ muscle with intact ACSL1 expression though the modulation of the content of signaling lipids, despite profound whole-body, HFD-induced IR. In our study HFD-fed mice presented all the aspects of IR at both the systemic and skeletal muscle level, together with muscular accumulation of bioactive DAG and Cer. Local Acsl1 silencing reverted detrimental effects of HFD feeding on muscular insulin sensitivity and glucose uptake. In order to explain the obtained results, we studied muscle lipid metabolism at several key levels. HFD feeding significantly increased the expression of fatty acids transporters and ACSL1 protein in HFD$_{(+Acsl1)}$ muscle, as reported previously [46]. Interestingly, the Acsl1 ablation concomitantly decreased the content of both ACSL1 and FATP1 protein in HFD$_{(-Acsl1)}$ gastrocnemius. In the study by Richards et al. FATP1 protein complexed with ACSL1 in 3T3-L1 adipocytes [47], which suggest that ACSL1 and FATP1 close interaction is needed for efficient channeling of extracellular FFA to acyl-CoA pool. In our study opposite effect was noted for FABPpm protein. This phenomenon could be the results of increased muscular insulin sensitivity in ACSL1-silenced gastrocnemius. Kawaguchi et al. noted that high glucose infusion rate in non-obese humans correlates with up-regulation of FABPpm and down-regulation of FATP1 in skeletal muscle [48]. Similar relationship was observed in skeletal muscle after 6 weeks of training [49]. High expression of CD36 and FABPpm in muscle of HFD-fed under ACSL1 down regulation raises the question of the fate of intramuscular free fatty acids. As acyl-CoA synthesis allows for both the intracellular capture of FFA and channeling towards metabolic pathways, non-esterified FA can either exit the cell through passive flip-flop mechanism [50,51] or bind to intracellular or membrane transport proteins [52]. As protein-mediated transport accounts for majority of FFA intracellular traffic [53], the upregulation of FABPpm in ACSL1-silenced muscle suggest the presence of compensatory buffer mechanism, which protects the cell from FFA lipotoxicity.

Regarding the mitochondrial channeling of fatty acids, we noted increased inhibition of ACC through its phosphorylation and increased expression of β-oxidation enzymes in HFD$_{(+Acsl1)}$ muscle. ACSL1 silencing in HFD$_{(-ACSL1)}$ muscle normalized both the CPT1B expression, ACC phosphorylation and decreased expression of β-oxidation proteins compared to non-silenced muscle. We also noted simultaneous reduction in the content of LCACoA as well as both the long- and short-chain acyl-carnitines which strongly suggests that ACSL1 down-regulation decreases skeletal muscle mitochondrial β-oxidation at the level of mitochondrial enzymes and acyl-CoA substrate aviability. Compensatory up-regulation of CPT1B, together with increased activation of ACC (through de-phosphorylation) can be interpreted as a possible response to rescue both the mitochondrial β-oxidation and de-novo lipogenesis in the conditions of decreased biological availability of acyl-CoAs. Acsl1 partial ablation could consequently normalize defects in mitochondrial β-oxidation observed under lipid overload and lead to balanced distribution of LCACoA towards β-oxidation and lipid synthesis. It has to be noted that observed modulation of particular LCACoA molecular species (e.g. C16:0 and C18:0 LCACoA) observed in non-silenced and silenced muscle is likely dependent on ACSL1 substrate specificity, which predominantly activates 16:0, 18:0 and 18:1 chain length FA to their acyl-CoAs [19]. Although significant inhibition of ACSL1 activity could lead to compensatory up-regulation of other ACSL isoforms, we did not observe this effect. Also the increase

in other isoform-specific acyl-CoAs (i.e. shorter than C16:0, or longer than C20:0 acyl-CoAs) was not observed, suggesting the predominant role of ACSL1 in muscular LCACoA production. Possible decrease in β-oxidation in ACSL1-silenced muscle rises question regarding utilization of other energy substrates. We observed increase insulin-stimulated glucose uptake and up-regulation of GLUT4 expression in HFD$_{(-Acsl1)}$ gastrocnemius, suggesting enhancement in carbohydrate utilization. Muscle specific ACLS1 knock-out mice indeed display decreased blood glucose and higher respiratory exchange ratio (RER) pointing towards carbohydrates as the predominant energy source [45]. Amino-acids could be the second preferred energy source in the state of LCACoA undersupply. The study by Zhao et al. had shown, that in muscle specific Acsl1 knock-out mice voluntary treadmill exercise increases amino-acid utilization and stimulates post-exercise protein synthesis, suggesting muscular protein breakdown [4]. Compared to both of the above studies which relied on complete Acsl1 ablation, shRNA-mediated down-regulation in our study rather corrects muscular lipid accumulation in HFD-fed mice than blocks LCACoA metabolism. Thus possible detrimental effects of Acsl1 ablation on energy metabolism and muscle physiology are less likely under its partial down-regulation than complete knock-out.

We hypothesized, that silencing of Acsl1 can improve muscular insulin sensitivity through inhibition of the synthesis of the bioactive lipids, namely DAG and Cer. DAG-dependent PKC activation promotes inhibitory serine/threonine phosphorylation and prevents tyrosine phosphorylation of the insulin receptor or IRS-1 which is crucial in the proper function of insulin signaling cascade [54]. In our study, modulation of ACSL1 expression by HFD feeding or silencing led to respective increase or decrease of DAG species containing 16:0, 18:0 and 18:1 fatty acids (C16:0/18:0, C18:0/18:0, C16:0/18:2, C18:0/18:2 and 18:0/20:0 DAGs), which suggests that DAG molecular species composition in muscle is strongly related to muscle LCA-CoA bioavailability and mimics Acsl1 substrate specificity. The changes in intramuscular DAG were concomitant with inhibitory IRS-1 phosphorylation at Ser1101. This particular site in IRS-1 molecule is known to relay inhibitory action of DAG on insulin signaling pathway through the activation of DAG-dependent PKC isoenzymes. Yu Li et al. [55] demonstrated that IRS-1 phosphorylation at Ser1101 through PKCθ is a critical in mediating FFA-induced IR in muscle cells. This also contributes to reduced tyrosine phosphorylation of IRS1 at Tyr632 and inhibition of the PI3K/Akt signaling. In our study HFD$_{(-Acsl1)}$ gastrocnemius displayed the decrease in specific DAG molecular species, augmentation of Tyr632 phosphorylation and inhibition of Ser1101phosphorylation of IRS-1 and up-regulation of PI3K kinase as compared to HFD$_{(+Acsl1)}$ muscle. This indicates, that down-regulation of Acsl1 enhances the action of the insulin pathway in DAG-dependent manner.

Previously published works demonstrate that Cer negatively affects the insulin pathway in skeletal muscle at the level of Akt/PKB, possibly through activation of protein phosphatase 2A-dependent de-phosphorylation [56]. In our study HFD-induced muscle IR in HFD$_{(+Acsl1)}$ gastrocnemius was accompanied by accumulation (among others) of C18:0-Cer and C24:0-Cer. In the case of ACSL1 down-regulation in HFD$_{(-Acsl1)}$ muscle the most considerable decrease was observed for C18:1-Cer and C24:1-Cer. Concomitantly, we observed a significant reduction of Akt and AS160 phosphorylation in HFD$_{(+Acsl1)}$ muscle, while in HFD$_{(-Acsl1)}$ muscle–an increase in Akt and AS160 phosphorylation. Those changes in Cer-dependent proteins of insulin signaling pathway indicate, that ACSL1 partial ablation normalizes muscular insulin signaling also at the level of ceramide signaling.

In our study Acsl1 down-regulation affected the DAG content in greater extent than that of Cer. Indeed, muscular DAG modulation through GPAT silencing was able to rescue skeletal muscle insulin sensitivity in HFD-fed mice [57]. Yet the presence of both mechanisms of DAG and Cer-induced inhibition of insulin pathway in muscle (via PKC-dependent and

PPA2-dependent mechanisms, respectively) cannot be excluded. Recent studies suggest that CERS1-synthetised C18:0 play a major role in HFD-induced skeletal muscle IR [17,18]. In our previous study, despite increase in muscular DAG concentration, HFD rats treated with ceramide synthesis inhibitor (myriocin), displayed significantly higher muscular insulin sensitivity than HFD-only counterparts [16]. Similarly, local SPT silencing in skeletal muscle improves insulin sensitivity despite elevated muscular DAG content [58]. This suggests greater importance of Cer in the induction of muscular IR, with particular emphasis on Cers1-derived C18:0- and C18:1-ceramide.

## Conclusion

Collectively, our work present, that the modulation of intracellular bioactive lipids through Acsl1 down-regulation can revert muscle specific deficiency in insulin action in HFD-induced muscular IR. Additionally, the results of this study show that localized, partial ablation of the Acsl1 gene normalizes function of insulin pathway despite the HFD-induced, whole-body IR. This points to the accumulation of Acsl1-derived bioactive lipids as the most important aspect of the induction of muscular IR. The inhibition of fatty acids activation through Acsl1 modulation could be a better therapeutic alternative in obese state than the targeting of individual bioactive lipid species, as it corrects the aspects of intramuscular lipid accumulation at both the Cer and DAG levels.

## Supporting information

**S1 Appendix. Additional materials and methods.**
(PDF)

**S1 Fig. Description of the experimental study design.** Study was performed on the gastrocnemius muscle of C57BL/6J mice.
(PDF)

**S2 Fig. Visualization of green fluorescent protein (TurboGFP) reporter gene expression in mouse hindlimb at 6 weeks after electroporation-mediated plasmid transfection.**
(PDF)

**S3 Fig. Characteristics of HFD-induced obesity and insulin resistance in C57BL/6J mice.**
(PDF)

**S4 Fig. Plasma glucose and 2-deoxy-[1,2-3H (N)]-D-glucose profiles during 0.5 U/kg intraperitoneal insulin challenge.**
(PDF)

**S5 Fig. The impact of Acsl1 silencing on the gene expression of other skeletal muscle acyl-CoA synthetase isoforms.**
(PDF)

**S1 Table. List of the antibodies used in the study.**
(PDF)

**S2 Table. Sequence of primes used in the study.**
(PDF)

**S3 Table. Plasma, blood morphology parameters and concentration of individual plasma free fatty acids of the LFD-fed and HFD-fed mice.**
(PDF)

**S4 Table. The effect of a high-fat diet on the content of individual lipid in mouse gastrocnemius.**
(PDF)

**S5 Table. The effect of in vivo shRNA-mediated Acsl1 gene down-regulation on the content of individual lipids in mouse gastrocnemius in high-fat diet mice.**
(PDF)

## Author Contributions

**Conceptualization:** Monika Imierska, Agnieszka Błachnio-Zabielska, Piotr Zabielski.

**Data curation:** Agnieszka Błachnio-Zabielska, Piotr Zabielski.

**Formal analysis:** Kamila Roszczyc-Owsiejczuk, Agnieszka Błachnio-Zabielska, Piotr Zabielski.

**Funding acquisition:** Agnieszka Błachnio-Zabielska.

**Investigation:** Kamila Roszczyc-Owsiejczuk, Emilia Sokołowska, Mariusz Kuźmicki, Karolina Pogodzińska, Agnieszka Błachnio-Zabielska.

**Methodology:** Agnieszka Błachnio-Zabielska, Piotr Zabielski.

**Supervision:** Agnieszka Błachnio-Zabielska.

**Writing – original draft:** Kamila Roszczyc-Owsiejczuk, Piotr Zabielski.

**Writing – review & editing:** Agnieszka Błachnio-Zabielska, Piotr Zabielski.

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
