## [Decision Letter · Decision Letter 0]

20 Dec 2023

PONE-D-23-38615shRNA-mediated down-regulation of Acsl1 reverses skeletal muscle insulin resistance in obese C57BL6/J micePLOS ONE

Dear Dr. Zabielski,

Thank you for submitting your manuscript to PLOS ONE. After careful consideration, we feel that it has merit but does not fully meet PLOS ONE’s publication criteria as it currently stands. Therefore, we invite you to submit a revised version of the manuscript that addresses the points raised during the review process.

We look forward to receiving your revised manuscript.

Kind regards,

Juan J Loor

Academic Editor

PLOS ONE

 [This research was funded by a Foundation for Polish Science, grant number TEAM/2016-1/2 and Medical University of Bialystok, grant numbers SUB/1/DN/20/003/1117 and SUB/1/DN/21/002/1204.].  

[This research was funded by a Foundation for Polish Science, grant number TEAM/2016-1/2 and Medical University of Bialystok, grant numbers SUB/1/DN/20/003/1117 and SUB/1/DN/21/002/1204.]

 [This research was funded by a Foundation for Polish Science, grant number TEAM/2016-1/2 and Medical University of Bialystok, grant numbers SUB/1/DN/20/003/1117 and SUB/1/DN/21/002/1204.].

6. We note that Figure S2 and S3 in your submission contain copyrighted images. All PLOS content is published under the Creative Commons Attribution License (CC BY 4.0), which means that the manuscript, images, and Supporting Information files will be freely available online, and any third party is permitted to access, download, copy, distribute, and use these materials in any way, even commercially, with proper attribution. For more information, see our copyright guidelines: http://journals.plos.org/plosone/s/licenses-and-copyright.

a. You may seek permission from the original copyright holder of Figure S2 and S3 to publish the content specifically under the CC BY 4.0 license. 

Reviewers' comments:

Reviewer's Responses to Questions

**Comments to the Author**

1. Is the manuscript technically sound, and do the data support the conclusions?

Reviewer #1: Partly

Reviewer #2: No

2. Has the statistical analysis been performed appropriately and rigorously? 

Reviewer #1: No

Reviewer #2: No

3. Have the authors made all data underlying the findings in their manuscript fully available?

Reviewer #1: Yes

Reviewer #2: Yes

4. Is the manuscript presented in an intelligible fashion and written in standard English?

Reviewer #1: Yes

Reviewer #2: Yes

5. Review Comments to the Author

Reviewer #1: Roszczyc-Owsiejczuk et al., investigated the role of ACSL1 in skeletal muscle in regulating insulin resistance under high fat diet using mouse model. It is an interesting topic. However, I have some concerns about the experiment design. My comments are listed below:

1. The introduction is too long. Please reorganize this part and make it more concise.

2. What is the age and sex of the mice used in this experiment, please clarify. Did you check the food intake of the mice? Did you check muscle mass and muscle histology for all the groups? If lipid metabolism has been affected, it is possible to see the difference of lipid droplet staining in skeletal muscle between -shRNA and +shRNA ACSL1 group under the HFD condition.

3. The most confusing part is the experiment design. Why did not have LFD -shRNA ACSL1 group? It is not clear how many groups you used in the current study based on your description in materials and methods. I believe you have 4 groups LFD, HFD and HFD-shRNA ACSL1, HFD+shRNA ACSL1 (Although the last two groups were actually left and right leg of the same mouse) based on your Figure S1. However, you did not mention HFD group in your materials and methods. At least to me, it is not reasonable to compare LFD with both legs treated with scrambled shRNA with only one leg treated with scrambled shRNA and the other leg treated with shRNA ACSL1. If in the LFD condition you also perform the same treatment like the HFD it is acceptable. Then it could be a 2x2 factorial experiment design.

4. Based on your results, high fat diet could increase the protein expression of ACSL1. Even though you knockdown ACSL1 but the protein level is still higher than LFD condition which suggests that diet may have a stronger effect than ACSL1 itself. That is also why it is necessary to knockdown ACSL1 in the LFD condition to see the efficiency of its knockdown. Besides, you noticed the insulin signaling pathway has been altered in ACSL1 knockdown condition, but overall systemic metabolism did not change based on your glucose tolerance test and insulin tolerance test. This also suggest that other organs such as liver or adipose tissue might be affected in the HFD condition.

5. It will be nice to do lipidomics using skeletal muscle tissue and plasma which can help understand the role of ACSL1 in skeletal muscle.

Reviewer #2: Summary

This paper demonstrates that ablation of the lipid-activating enzyme Acsl1 in skeletal muscle is beneficial and ameliorates insulin resistance. The authors utilize single leg electroporation to deliver plasmids against Acsl1 to one leg and show that, in a high fat-diet context, inhibition of the fatty acid conversion to the acyl CoA-derivatives is able to rescue the insulin resistant phenotype observed in the contralateral leg muscle. This is relevant, given that the uptake of free fatty acids is not reduced, but without activation into acyl-CoA, lipid synthesis byproducts such as DAG and Sphingolipids are not produced. This is confirmed by a reversal of the insulin resistant phenotype. Altogether, Acsl1 ablation is sufficient to rescue the insulin resistant muscle.

Major comments

The paper is clear and the phenotype reported is relevant to understand how free fatty acid activation by Acsl1 is at the crossroads of fatty acid oxidation and lipid synthesis in muscle. My major concern with the current paper is the statistical analysis. As described in the methods section (Page 6, line 128; Supplement Figure S2), the transfection of the Acsl1 snRNA was done in the “Left gastrocnemius muscle of high-fat diet mice was electroporated with active shRNA plasmid… yielding silenced HFD(-Acsl1) gastrocnemius, while the contralateral, right hindlimb gastrocnemius within the same animal was transfected with scrambled shRNA, yielding HFD(+Acsl1) gastrocnemius.” Additionally, the authors provide a schematic to describe the experimental design: 16 mice were used for the transfection studies, where only one group (n=8) of HFD-fed mice were transfected on the each leg with different shRNAs. Thus, this presents difficulties when performing statistical tests. Given that only HFD mice, and not LFD, have a left (-Acsl1- gasctrocnemius, comparing to LFD is not a valid or direct control. This is because the LFD group received scrambled shRNA on both legs, and no Acsl1- shRNA. Thus, the group comparison is not possible. Additionally, the authors treat the HFD-fed legs as different and independent groups (showing 3 groups total). It is misleading to display data in this way. Investigators should be discouraged from visualizing within animal and between animal comparisons on the same figure. These data could be compared such as with paired statistics or two separate unpaired tests, all with correction for Family wise error. Additionally, tests should be corrected for multiplicity using Bonferroni or similar correction factor. This is important, given that previous papers that are cited for single leg transfections make use of such statistics. This treatment has systemic effects, and thus the data from left leg and right leg are dependent upon each other, and should be a paired. Investigators should re-run statistics and adjust all figures and tables to avoid comparison between each leg of the same mouse and across groups within the same graph.

Given only two groups are examined in the current paper, how were the metabolic tests performed to show 3 groups?

The authors demonstrate that acyl-CoAs, diacylglycerides, and sphingolipids are reduced in the Acsl1- muscle, and further show that this is followed by a paradoxical increase in CPT1B, but a reduction in��-oxidation proteins (HADHA, ACADVL, ACADL) and acyl-carnitines. However, no changes in FFA are observed between the HFD left Acsl1- and right Acsl1+ gastrocnemius muscles or fatty acid transporters. What is the fate of these FFA? What is the interpretation for such results? This should be added to the discussion.

Could the increase in the right Acsl1+ gastrocnemius be a compensation for reduced fatty acid activation by the left Acsl1- muscle? Do other muscles experience this? Are the changes observed in the Acsl1+ gastrocnemius similar to a HFD with both Acsl1+ legs?

The beneficial effects of Acsl- ablation are important and increase our understanding of how fatty acid activation can be utilized to restore insulin sensitivity to the muscle. What is the effect on energy metabolism? Other cited papers show that these muscles increase the use of glucose and amino acids. This should be expanded with the GLUT4 and glucose uptake findings.

6. PLOS authors have the option to publish the peer review history of their article (what does this mean?). If published, this will include your full peer review and any attached files.

Reviewer #1: **Yes: **Yusheng Liang

Reviewer #2: **Yes: **Diego Hernandez-Saavedra

---

## [Author Response · Author response to Decision Letter 0]

30 Jan 2024

Reviewer #1: Roszczyc-Owsiejczuk et al., investigated the role of ACSL1 in skeletal muscle in regulating insulin resistance under high fat diet using mouse model. It is an interesting topic. However, I have some concerns about the experiment design. My comments are listed below:

Reply: We thank the Reviewer for his effort to improve our manuscript. We hope, that updated manuscript will gain the Reviewer recommendation for publication.

1. The introduction is too long. Please reorganize this part and make it more concise.

Reply: To improve its readability, we shortened and re-organized the introduction as requested. 

2. What is the age and sex of the mice used in this experiment, please clarify. Did you check the food intake of the mice? Did you check muscle mass and muscle histology for all the groups? If lipid metabolism has been affected, it is possible to see the difference of lipid droplet staining in skeletal muscle between -shRNA and +shRNA ACSL1 group under the HFD condition.

Reply: The study was performed on male C57BL/6J mice of 6 weeks of age at the beginning of the experiment. Mice were fed for 8 weeks with HFD to induce whole-body insulin resistance. Animals were euthanized at the age of 14-15 weeks. Above information was previously included in “Materials and Methods” section. We introduced “Animals” paragraph and put this information at the beginning of the “Methods” secretion, as previously it was somehow hidden in the experiment design details. 

Mice were housed at 4 individuals per cage, according to IACUC-accepted procedure. Unfortunately, individual food consumption monitoring was not possible.

The weight of non-silenced muscle and ACSL1-silenced muscle in HFD-fed animals was indifferent between both legs and presented physiological morphology (Supplement Figure S2). During our initial small-scale experiment to refine electroporation technique, the muscle displayed normal histology under HE (hematoxylin/eosin) stain, except small necrotic area directly around needle injection. This was observed similarly in both non-silenced and Acsl1-silenced muscle.

Regarding lipid droplet stain, unfortunately we did not perform appropriate preparations, due to the fact, that all the gastrocnemius muscle tissues from experimental animals were pulverized in LN2 prior to all the subsequent measurements. Nevertheless, the drop in muscle TAG accumulation (now shown in Figure 3A2) in HFD(-ACSL1) gastrocnemius as compared to HFD(+Acsl1) gastrocnemius can be a good indicator of possible decrease in lipid droplet content in Acsl1 down-regulated muscle.

3. The most confusing part is the experiment design. Why did not have LFD -shRNA ACSL1 group? It is not clear how many groups you used in the current study based on your description in materials and methods. I believe you have 4 groups LFD, HFD and HFD-shRNA ACSL1, HFD+shRNA ACSL1 (Although the last two groups were actually left and right leg of the same mouse) based on your Figure S1. However, you did not mention HFD group in your materials and methods. At least to me, it is not reasonable to compare LFD with both legs treated with scrambled shRNA with only one leg treated with scrambled shRNA and the other leg treated with shRNA ACSL1. If in the LFD condition you also perform the same treatment like the HFD it is acceptable. Then it could be a 2x2 factorial experiment design.

Reply: Thank you for this valuable comment. We agree that performing statistical analysis between those 3 tissues (LFD(+Acsl1), HFD(+Acsl1) and HFD(-Acsl1) gastrocnemius) using ANOVA statistics was not optimal. We changed both the data presentation and statistical analysis. In our study, non-silenced gastrocnemius from LFD-fed mice (LFD(+Acsl1) muscle) served as a baseline, insulin-sensitive tissue for non-silenced insulin-resistant gastrocnemius form HFD-fed mice (HFD(+Acls1) muscle). In the revised manuscript we present this comparison on separate figures and use non-parametric Wilcoxon rank sum test for unpaired samples to establish significance. For within-animal comparison between non-silenced HFD(+Acls1) muscle and contralateral silenced HFD(-Acsl1) muscle, we employed non-parametric Wilcoxon signed rank test for paired samples. We also present the results of this comparison on separate figures. In both unpaired- and paired comparisons we opted for non-parametric Wilcoxon test over respective t-test due to small sample size. Although comparisons using Wilcoxon statistics gave higher p-values than t-test, significance was retained. We also modified the results section to clarify the confusion regarding the experimental groups and their comparisons.

Our rationale to obtain both non-silenced and silenced gastrocnemius from the same animal with HFD-induced insulin resistance was to exclude the influence of other factors, which could affect skeletal muscle insulin resistance. Thus the differences in gastrocnemius insulin action and bioactive lipid content between ACSL1 silenced (HFD(-Acsl1) and non-silenced (HFD(+Acsl1) muscle can be only attributed to ACSL1 modulation. The non-silenced LFD(+Acsl1) muscle was introduced as a baseline, insulin sensitive tissue, to observe the effects of HFD feeding on muscle lipid metabolism and insulin sensitivity in silenced HFD(+Acsl1) muscle. 

Moreover, we updated Figure S2 to better present our experimental design.

We hope that updated manuscript fully addresses those important issues.

4. Based on your results, high fat diet could increase the protein expression of ACSL1. Even though you knockdown ACSL1 but the protein level is still higher than LFD condition which suggests that diet may have a stronger effect than ACSL1 itself. That is also why it is necessary to knockdown ACSL1 in the LFD condition to see the efficiency of its knockdown. Besides, you noticed the insulin signaling pathway has been altered in ACSL1 knockdown condition, but overall systemic metabolism did not change based on your glucose tolerance test and insulin tolerance test. This also suggest that other organs such as liver or adipose tissue might be affected in the HFD condition.

Reply: We are grateful for this important remark. In our experiment we achieved modest down-regulation of Acsl1 at protein level. Much stronger effect was achieved at mRNA expression level. We hypothesize, that due to very slow turnover rate of muscle proteins, the silenced gastrocnemius could still retain Acsl1 protein from pre-electroporation stage, yet with decreased biological activity. This is evidenced by significant drop in LCACoA content, most visible for 16- and 18-carbon CoAs (preferred Acsl1 substrate/product fatty acids chain length). 

Regarding the systemic effects, we chosen local single gastrocnemius Acsl1 silencing in HFD-fed insulin resistant mice especially not to affect whole-body insulin resistance. The affected muscle volume was too small to modify systemic FA metabolism. Thus all the beneficial effects on muscle insulin sensitivity could be attributed solely to the local Acsl1 silencing, not to the modification of other factors. We ruled-out Acsl1 knock-out approach, as it modifies whole-body FA metabolism blurring the muscle-specific mechanisms of insulin resistance. In the Materials for Reviewers, we present Figure S1 and Table S1, which compares anthropometric values of HFD animals with no silencing (from our initial experiments, HFD1 animals) with values of HFD animals with single gastrocnemius silencing (HFD(-Acsl1)). Data indicates, that local Acsl1 silencing had no observable effects at systemic level. 

We agree, that other organs e.g. liver and adipose tissue are crucial in the multifaceted mechanisms of systemic insulin resistance. Our experiment aimed at elucidation of muscle-specific mechanisms of diet induced IR. We also performed a follow-up project, which employed liver-specific shRNA-mediated silencing of lipid-metabolism enzymes via hydrodynamic gene delivery technique, which results are currently being prepared for publication.

5. It will be nice to do lipidomics using skeletal muscle tissue and plasma which can help understand the role of ACSL1 in skeletal muscle.

Reply: The non-targeted lipidomic approach combined with stable isotope labeling and LC/MS/MS is a powerful tool which could shed the light on the fate of fatty acids on Acsl1-silenced tissue. Regarding the current study, we had no access to this methodology. Nevertheless, our targeted LC/MS/MS-based technique cover significant portion of bioactive lipidome, especially molecular species of DAG and Cer with proven biological activity. 

Reviewer #2: Summary

This paper demonstrates that ablation of the lipid-activating enzyme Acsl1 in skeletal muscle is beneficial and ameliorates insulin resistance. The authors utilize single leg electroporation to deliver plasmids against Acsl1 to one leg and show that, in a high fat-diet context, inhibition of the fatty acid conversion to the acyl CoA-derivatives is able to rescue the insulin resistant phenotype observed in the contralateral leg muscle. This is relevant, given that the uptake of free fatty acids is not reduced, but without activation into acyl-CoA, lipid synthesis byproducts such as DAG and Sphingolipids are not produced. This is confirmed by a reversal of the insulin resistant phenotype. Altogether, Acsl1 ablation is sufficient to rescue the insulin resistant muscle.

Reply: We are grateful for the valuable comments by Reviewer, which allowed us to improve our manuscript. We hope, that employed corrections will answer all the important questions raised by the Reviewer and fulfill the standards for the recommendation for publication.

Major comments

The paper is clear and the phenotype reported is relevant to understand how free fatty acid activation by Acsl1 is at the crossroads of fatty acid oxidation and lipid synthesis in muscle. My major concern with the current paper is the statistical analysis. As described in the methods section (Page 6, line 128; Supplement Figure S2), the transfection of the Acsl1 snRNA was done in the “Left gastrocnemius muscle of high-fat diet mice was electroporated with active shRNA plasmid… yielding silenced HFD(-Acsl1) gastrocnemius, while the contralateral, right hindlimb gastrocnemius within the same animal was transfected with scrambled shRNA, yielding HFD(+Acsl1) gastrocnemius.” Additionally, the authors provide a schematic to describe the experimental design: 16 mice were used for the transfection studies, where only one group (n=8) of HFD-fed mice were transfected on the each leg with different shRNAs. Thus, this presents difficulties when performing statistical tests. Given that only HFD mice, and not LFD, have a left (-Acsl1- gasctrocnemius, comparing to LFD is not a valid or direct control. This is because the LFD group received scrambled shRNA on both legs, and no Acsl1- shRNA. Thus, the group comparison is not possible. Additionally, the authors treat the HFD-fed legs as different and independent groups (showing 3 groups total). It is misleading to display data in this way. Investigators should be discouraged from visualizing within animal and between animal comparisons on the same figure. These data could be compared such as with paired statistics or two separate unpaired tests, all with correction for Family wise error. Additionally, tests should be corrected for multiplicity using Bonferroni or similar correction factor. This is important, given that previous papers that are cited for single leg transfections make use of such statistics. This treatment has systemic effects, and thus the data from left leg and right leg are dependent upon each other, and should be a paired. Investigators should re-run statistics and adjust all figures and tables to avoid comparison between each leg of the same mouse and across groups within the same graph.

Reply: We are grateful for pointing us towards this issue. We agree that performing statistical analysis between 3 tissues (LFD+, HFD+ and HFD- gastrocnemius) using ANOVA statistics was not optimal. According to the Reviewer comments, we changed both the data presentation and statistical analysis. In the updated manuscript, significance between gastrocnemius from LFD-fed animals (sham-electroporated LFD(+Acsl1) muscle) and matching gastrocnemius from HFD-fed animals (sham-electroporated HFD(+Acsl1) muscle) was tested with non-parametric Wilcoxon rank sum test for unpaired samples and the data is presented on separate figure panels. For within-animal comparison between sham-electroporated HFD(+Acsl1) muscle and silenced HFD(-Acsl1) muscle, we employed non-parametric Wilcoxon signed rank test for paired samples and similarly, we present the results on separate figures. In both unpaired- and paired comparisons we opted for non-parametric Wilcoxon test over respective t-test due to our small sample size and higher power of the Wilcoxon test. Although comparisons using Wilcoxon statistics gave higher p-values than that of t-test (or our original ANOVA), significance was retained. We also modified the results section and updated Figure S2 to clarify the confusion regarding the experimental groups. 

We hope that updated manuscript fully addresses those important issues.

Given only two groups are examined in the current paper, how were the metabolic tests performed to show 3 groups?

Reply: We understand that the Reviewer is pointing us towards HFD1 animals from original Supplement File S1 (anthropometric and metabolic tests). This particular group of HFD-fed animals had gastrocnemius in both legs sham-electroporated with non-coding plasmid. We measured only anthropometric/metabolic parameters in those animals. Anthropometric values from this group show, that there is no systemic alternations between those animals and the animals with Acsl1 down regulation in single gastrocnemius (HFD group which yields both +Acsl1 and -Acsl1 tissue). This is expected, as down-regulation of Acsl1 in single gastrocnemius is unlikely to exert noticeable effects at systemic level (e.g plasma fatty acids). Contrary to pharmaceutical intervention, whole-body knock out or muscle-specific knock-out, local silencing has no systemic effects. Thus the improvement in muscle-level insulin action in Acsl1-down regulated muscle can be attributed only to the modulation of Acsl1 and its LCACoA product but no other systemic factors.

We are sorry, that introduction of this particular set of animals led to misunderstanding. As introduction of those animals led to confusion, we removed the data from HFD1 animals from the revised manuscript and Supplement File 1. 

The authors demonstrate that acyl-CoAs, diacylglycerides, and sphingolipids are reduced in the Acsl1- muscle, and further show that this is followed by a paradoxical increase in CPT1B, but a reduction in��-oxidation proteins (HADHA, ACADVL, ACADL) and acyl-carnitines. However, no changes in FFA are observed between the HFD left Acsl1- and right Acsl1+ gastrocnemius muscles or fatty acid transporters. What is the fate of these FFA? What is the interpretation for such results? This should be added to the discussion.

Reply: We did not analyze intracellular free fatty acids in the muscle tissue. The FFA data from previous version of Supplement Table S1 (now in Materials for Reviewers) refers to the systemic parameters, e.g plasma free fatty acids. Regarding the fate of intramuscular fatty acids in Acsl1-silenced tissue, we hypothesize, that since FFA activation to Acyl-CoA is decreased, excess of plasma FFA do not enter metabolic pathways which lead to lipid synthesis or β-oxidation. Decrease in muscle mitochondrial β-oxidation was observed in the muscle of ACSL1 knock out mice fed low-fat diet. As both Acsl1 and fatty acids transporter FATP1 co-localize and create complex in plasma membrane and mitochondrial outer membrane, mitochondrial Acyl-CoA uptake could be significantly decreased, as evidenced by significant decrease in acyl-carnitines in Acsl1-silenced muscle. The decrease in the Acyl-CoA flow through metabolic pathways could be the reason for paradoxical up-regulation of both the FAPBpm membrane fatty acids transporter and CPT1 protein. Yet with decreased metabolic capture of FFA into LCACoA, free fatty acids could diffuse ou

---

## [Decision Letter · Decision Letter 1]

6 May 2024

PONE-D-23-38615R1shRNA-mediated down-regulation of Acsl1 reverses skeletal muscle insulin resistance in obese C57BL6/J micePLOS ONE

Dear Dr. Zabielski,

Thank you for submitting your manuscript to PLOS ONE. After careful consideration, we feel that it has merit but does not fully meet PLOS ONE’s publication criteria as it currently stands. Therefore, we invite you to submit a revised version of the manuscript that addresses the points raised during the review process. **Both referee find our work interesting but have some concerns about the analysis of some data. Please revised the data presented in the Figure1. It will also be interesting to look at phosphorylation of Akt on Thr308 as suggested by the second referee.** Please submit your revised manuscript by Jun 20 2024 11:59PM. If you will need more time than this to complete your revisions, please reply to this message or contact the journal office at plosone@plos.org. Please include the following items when submitting your revised manuscript:A rebuttal letter that responds to each point raised by the academic editor and reviewer(s). You should upload this letter as a separate file labeled 'Response to Reviewers'.A marked-up copy of your manuscript that highlights changes made to the original version. You should upload this as a separate file labeled 'Revised Manuscript with Track Changes'.An unmarked version of your revised paper without tracked changes. You should upload this as a separate file labeled 'Manuscript'.

We look forward to receiving your revised manuscript.

Kind regards,

Herve Le Stunff

Academic Editor

PLOS ONE

Reviewers' comments:

Reviewer's Responses to Questions

**Comments to the Author**

1. If the authors have adequately addressed your comments raised in a previous round of review and you feel that this manuscript is now acceptable for publication, you may indicate that here to bypass the “Comments to the Author” section, enter your conflict of interest statement in the “Confidential to Editor” section, and submit your "Accept" recommendation.

Reviewer #1: (No Response)

Reviewer #2: All comments have been addressed

Reviewer #3: (No Response)

Reviewer #4: (No Response)

2. Is the manuscript technically sound, and do the data support the conclusions?

Reviewer #1: No

Reviewer #2: Yes

Reviewer #3: Partly

Reviewer #4: Yes

3. Has the statistical analysis been performed appropriately and rigorously? 

Reviewer #1: I Don't Know

Reviewer #2: Yes

Reviewer #3: No

Reviewer #4: Yes

4. Have the authors made all data underlying the findings in their manuscript fully available?

Reviewer #1: Yes

Reviewer #2: Yes

Reviewer #3: Yes

Reviewer #4: Yes

5. Is the manuscript presented in an intelligible fashion and written in standard English?

Reviewer #1: Yes

Reviewer #2: Yes

Reviewer #3: Yes

Reviewer #4: Yes

6. Review Comments to the Author

**Reviewer #1: **(No Response)

**Reviewer #2:** (No Response)

**Reviewer #3: **“shRNA-mediated down-regulation of Acsl1 reverses skeletal muscle insulin resistance in obese C57BL6/J mice”, by Roszczyc-Owsiejczuk et al.

The aim of this study was to determine whether long-chain acyl-CoA (LCACoA) and the enzyme that synthesises them, long-chain acyl-CoA synthetase (Acsl1), can modulate muscle insulin sensitivity in mice. The authors KO'd Acsl1 by electroporating a shRNA directed against Acsl1 into the gastrocnemius muscle of mice, while the muscle of the other leg was electroporated with a control shRNA. The animals were then fed a fatty diet for 8 weeks.

The authors show that reduced expression of ACSL1 induces a reduction in LCACoA, ceramides and diacylglycerols, and improves insulin sensitivity compared with control muscle. They also observed a reduction in mitochondrial fatty acid metabolism.

The results are interesting and show the involvement of long-chain fatty acids in the development of muscle insulin resistance in mice.

However, a number of important details raise questions and need to be clarified for the data to be unequivocal.

An important detail in the introduction to the article: the authors state that ceramides inhibit the muscular insulin response by activating PP2A phosphatase, which acts negatively on Akt (line 62-63). However, it has been clearly shown previously that in muscle cells ceramides act via activation of PKCz (Powell et al, 2003, Fox et al, 2007, Hajduch et al, 2008, Mahfouz et al, 2014). The authors cannot ignore this fact.

The major problem with the article is in Figure 1, where the authors show that electroporation worked and that shRNA decreased Acsl1 expression in the gastrocnemius of mice. However, not everything seems very clear. In Figure 1B2, the authors show a decrease in Acsl1 expression in the muscle in response to shRNA. This decrease appears to be statistically real. They also state that they did not observe any compensation with the other Acsl isoforms (Supplementary Figure 5). However, when we look at this last figure, the results do not seem so clear-cut. Indeed, the error bars are often larger than the expression levels of the isoforms. As a result, of course, there is no statistical difference between electroporated and non-electroporated. I don't understand how there can be so much variation between mice (8 mice) concerning the expression of these Acsl isoforms and, on the contrary, so little concerning Acsl1 (figure 1B2).

I think it's necessary for the authors to redo their PCRs because it's obvious that something isn't right.

Another major problem concerns the blots shown in all the figures. The authors have cut the bands they were interested in and put them side by side in the figures because the original blots contain many unused bands. This doesn't seem right to me. First of all, I'd like to know what all these unused strips in all the blots correspond to. What's more, for each point, there are 4 strips in the uncut blots. However, the authors only show 2 in the final figures. How did they choose the final bands? For the quantifications, did they quantify the 4 bands or the two bands? This is a real problem for me. Especially concerning the Acsl1 bands. It seems to me that the contrast has been changed between the two bands corresponding to HFD(+Acsl1) presented in figure 1 and the same bands in the original blot.

For me, the authors should migrate again their samples side by side so that they can present them properly. I must admit that as it stands, there may remain some doubt as to the value of the data.

Another point to raise, are you sure the data is statistically different in Figure 1B1? Looking at the error bars, I have my doubts. Is Acsl1 expression indeed increased in muscle in response to HFD? The authors should show the dispersion of points in all their results so that variations between experimental conditions can be seen.

I find the data in the paper interesting but their importance depends on the results of Figure 1 which poses a problem for me.

**Reviewer #4: **In this article, the authors explore the impact of ACSL1 on insulin resistance in skeletal muscle within the context of diet-induced obesity (DIO) in C57Bl6/J mice. Despite the unclear connection between ACSL1, fatty acid beta oxydation, and lipid metabolism concerning ectopic lipid accumulation, the authors employed electroporation to deliver a shRNA plasmid targeting ACSL1 to one gastrocnemius muscle while the other gastrocnemius received a sham shRNA plasmid.

The authors highlight modifications in ACSL1 expression, fatty acid beta-oxydation proteins, bioactive lipids contents, and insulin signaling during DIO. They demonstrate that inhibiting the conversion of fatty acids into SCA-coa and LCA-coa using an shRNA plasmid targeting ACSL1 can mitigate the harmful effects of DIO. Specifically, they report normalization of ACSL1 expression, restoration of SCA-coa and LCA-coa, DAG, and cermides levels, as well as the recovery of beta-oxidation protein expression in their experimental conditions.

Major comments:

1. The authors chose to carry out their study using the C57Bl6/J mouse strain instead of C57Bl6/N. It is essential to justify this decision considering the references below:

o Kaku K, Fiedorek FT Jr, Province M, Permutt MA: Genetic analysis of glucose tolerance in inbred mouse strains: evidence for polygenic control. Diabetes 37:707–713, 1988.

o Kooptiwut S, Zraika S, Thorburn AW, Dunlop ME, Darwiche R, Kay TW, Proietto J, Andrikopoulos S: Comparison of insulin secretory function in two mouse models with different susceptibility to beta-cell failure. Endocrinology 143:2085–2092, 2002.

o Freeman HC, Hugill A, Dear NT, Ashcroft FM, Cox RD: Deletion of nicotinamide nucleotide transhydrogenase: a new quantitative trait locus accounting for glucose intolerance in C57BL/6J mice. Diabetes. 2006 Jul;55(7):2153-6.

The authors’ DIO effectively induced glucose intolerance and insulin resistance, as evidenced by comparisons with the LFD group, and their selection of C57Bl6/J strain does not undermine the study’s validity.

2. The authors do not specify whether CPT2 is altered under their experimental conditions or if a compensatory mechanism occurs in HFD(+Acsl1) and HFD(-Acsl1).

3. In Figure 4, the authors measure the phosphorylation of Akt/PKB by assessing the phosphorylation of Akt Ser473 using western blotting. It would be valuable to know if they also conducted western blot analysis with an antibody targeting Thr308-Akt since pAKT/PKB requires activation on both sites (Thr308 and Ser473) for full signaling pathway activity.

Minor comments:

1. The figures in the revised submission appear to have formatting issues, making them difficult to read, particularly Figures 2 and 4. It is crucial to upload them in higher resolution.

2. The references’ section (line 422) appears to use two different formats for presenting references. Kindly standardize the format.

3. In the introduction and discussion, the authors inconsistently describe lipid species using either the “C16:0” notation or “18-carbon Cer.” A more consistent presentation would improve readability.

7. PLOS authors have the option to publish the peer review history of their article (what does this mean?). If published, this will include your full peer review and any attached files.

Reviewer #1: **Yes: **Yusheng Liang

Reviewer #2: No

Reviewer #3: No

Reviewer #4: **Yes: **Cécile L. Bandet

---

## [Author Response · Author response to Decision Letter 1]

21 Jun 2024

Manuscript: PONE-D-23-38615R1

shRNA-mediated down-regulation of Acsl1 reverses skeletal muscle insulin resistance in obese C57BL6/J mice

Response to Reviewers

Dear Herve Le Stunff,

Thank You for giving us the opportunity to submit a revised version of the manuscript for possible publication in the PLOS ONE. We appreciate the time and effort that You and the Reviewers have dedicated to providing the feedback on our manuscript. We are grateful for the insightful comments and valuable improvements to our paper. We have incorporated both the Editor, Editorial Office and the Reviewers' suggestions. Please find below our point-by-point response to the Reviewers’ comments and concerns.

Response to Reviewer 1:

Comment from Reviewer 1:The aim of this study was to determine whether long-chain acyl-CoA (LCACoA) and the enzyme that synthesises them, long-chain acyl-CoA synthetase (Acsl1), can modulate muscle insulin sensitivity in mice. The authors KO'd Acsl1 by electroporating a shRNA directed against Acsl1 into the gastrocnemius muscle of mice, while the muscle of the other leg was electroporated with a control shRNA. The animals were then fed a fatty diet for 8 weeks. The authors show that reduced expression of ACSL1 induces a reduction in LCACoA, ceramides and diacylglycerols, and improves insulin sensitivity compared with control muscle. They also observed a reduction in mitochondrial fatty acid metabolism. The results are interesting and show the involvement of long-chain fatty acids in the development of muscle insulin resistance in mice. However, a number of important details raise questions and need to be clarified for the data to be unequivocal.

Author’s response: We would like to thank the Reviewer for providing valuable comments that have allowed us to improve our manuscript. Please find below our detailed response.

Comment from Reviewer 1:An important detail in the introduction to the article: the authors state that ceramides inhibit the muscular insulin response by activating PP2A phosphatase, which acts negatively on Akt (line 62-63). However, it has been clearly shown previously that in muscle cells ceramides act via activation of PKCz (Powell et al, 2003, Fox et al, 2007, Hajduch et al, 2008, Mahfouz et al, 2014). The authors cannot ignore this fact.

Author’s response: Thank You for bringing this to our attention. We have carefully revisited the Introduction section of the manuscript, and made the necessary revisions to address Reviewers concern regarding ceramide-mediated PKCζ activation.

Comment from Reviewer 1:The major problem with the article is in Figure 1, where the authors show that electroporation worked and that shRNA decreased Acsl1 expression in the gastrocnemius of mice. However, not everything seems very clear. In Figure 1B2, the authors show a decrease in Acsl1 expression in the muscle in response to shRNA. This decrease appears to be statistically real. They also state that they did not observe any compensation with the other Acsl isoforms (Supplementary Figure 5). However, when we look at this last figure, the results do not seem so clear-cut. Indeed, the error bars are often larger than the expression levels of the isoforms. As a result, of course, there is no statistical difference between electroporated and non-electroporated. I don't understand how there can be so much variation between mice (8 mice) concerning the expression of these Acsl isoforms and, on the contrary, so little concerning Acsl1 (figure 1B2). I think it's necessary for the authors to redo their PCRs because it's obvious that something isn't right.

Author’s response: We agree that variability in mRNA expression analysis of other Acsl isoforms was indeed excessive. As recommended, we re-analyzed all Acsl isoforms gene expression (from Acsl1 to Acsl6) and the data is presented in updated Figure 1 and Supplement 1 Figure S5. The new PCR measurements were performed in a single batch by the same technician. 

Comment from Reviewer 1:Another major problem concerns the blots shown in all the figures. The authors have cut the bands they were interested in and put them side by side in the figures because the original blots contain many unused bands. This doesn't seem right to me. First of all, I'd like to know what all these unused strips in all the blots correspond to. What's more, for each point, there are 4 strips in the uncut blots. However, the authors only show 2 in the final figures. How did they choose the final bands? For the quantifications, did they quantify the 4 bands or the two bands? This is a real problem for me. Especially concerning the Acsl1 bands. It seems to me that the contrast has been changed between the two bands corresponding to HFD(+Acsl1) presented in figure 1 and the same bands in the original blot. For me, the authors should migrate again their samples side by side so that they can present them properly. I must admit that as it stands, there may remain some doubt as to the value of the data.

Author’s response: Thank You for pointing us towards this inconsistency. In response to Your concerns and to ensure the accuracy of our data presentation, we conducted additional WB analysis for the Acsl1 protein and have included the results in Figure 1 B1-B2. Regarding additional lanes in the original blots, those belong to other experimental groups from the larger study. Those included muscle samples with silencing of other lipid metabolism-related genes, such as CerS or GPAT (data already published). Our desire was to perform the exemplary WBs for all the experimental groups in the single, large, 26-lane blot, thus reducing blot-to-blot variability. The updated Supplement 2 shows which samples were taken for Acsl1-related study.

Comment from Reviewer 1:Another point to raise, are you sure the data is statistically different in Figure 1B1? Looking at the error bars, I have my doubts. Is Acsl1 expression indeed increased in muscle in response to HFD? The authors should show the dispersion of points in all their results so that variations between experimental conditions can be seen. I find the data in the paper interesting but their importance depends on the results of Figure 1 which poses a problem for me.

Author’s response: Thank You for Your feedback regarding the presentation of our data. Regarding the Acsl1 expression at both the protein and mRNA levels, the Figure 1 now contains the data from re-analyzed samples (as mentioned earlier). Updated figures include both the medians (as bars), interquartile ranges (as whiskers) and individual data points (as appropriate markers). Additionally, we re-analyzed all the data using Student’s t-statistics in both the unpaired (for LFD(+Acsl1) vs HFD(+Acsl1) comparison) and paired (for HFD(+Acsl1) vs HFD(-Acsl1) comparison) versions (not shown in the manuscript) to confirm the findings. We noted similar, or - in most cases - higher significance results (smaller p-values) in the case of Student’s t-test than those observed in the case of Wilcoxon non-parametric test. Only in the case of SCA-Car values from LFD(+Acsl1) vs HFD(+Acsl1) comparison, the parametric t-test gave non-significant results as compared to Wilcoxon test. Although individual data-point distribution indeed suggests lack of significance in the abovementioned comparison, we believe that the use of non-parametric statistics is more applicable for our study due to small n-numbers which do not allow for proper testing of data normality, and thus exclude employment of parametric tests such as paired or un-paired Students t-test. We are hesitant to selectively present t-test significance results for SCA-Car data with other measurements tested by the Wilcoxon method, unless the Reviewer will insist otherwise. We hope that the updated data presentation will sufficiently address the Reviewer questions.

Reviewer #4: In this article, the authors explore the impact of ACSL1 on insulin resistance in skeletal muscle within the context of diet-induced obesity (DIO) in C57Bl6/J mice. Despite the unclear connection between ACSL1, fatty acid beta oxydation, and lipid metabolism concerning ectopic lipid accumulation, the authors employed electroporation to deliver a shRNA plasmid targeting ACSL1 to one gastrocnemius muscle while the other gastrocnemius received a sham shRNA plasmid.

The authors highlight modifications in ACSL1 expression, fatty acid beta-oxydation proteins, bioactive lipids contents, and insulin signaling during DIO. They demonstrate that inhibiting the conversion of fatty acids into SCA-coa and LCA-coa using an shRNA plasmid targeting ACSL1 can mitigate the harmful effects of DIO. Specifically, they report normalization of ACSL1 expression, restoration of SCA-coa and LCA-coa, DAG, and cermides levels, as well as the recovery of beta-oxidation protein expression in their experimental conditions.

Major comments:

1. The authors chose to carry out their study using the C57Bl6/J mouse strain instead of C57Bl6/N. It is essential to justify this decision considering the references below:

o Kaku K, Fiedorek FT Jr, Province M, Permutt MA: Genetic analysis of glucose tolerance in inbred mouse strains: evidence for polygenic control. Diabetes 37:707–713, 1988.1

o Kooptiwut S, Zraika S, Thorburn AW, Dunlop ME, Darwiche R, Kay TW, Proietto J, Andrikopoulos S: Comparison of insulin secretory function in two mouse models with different susceptibility to beta-cell failure. Endocrinology 143:2085–2092, 2002. 

o Freeman HC, Hugill A, Dear NT, Ashcroft FM, Cox RD: Deletion of nicotinamide nucleotide transhydrogenase: a new quantitative trait locus accounting for glucose intolerance in C57BL/6J mice. Diabetes. 2006 Jul;55(7):2153-6. 

The authors’ DIO effectively induced glucose intolerance and insulin resistance, as evidenced by comparisons with the LFD group, and their selection of C57Bl6/J strain does not undermine the study’s validity.

Author’s response: 

We acknowledge the significance of the provided references, which document differences in both the genotype and phenotype between different mice strains. We agree that C57BL/6J-specific mutation in Nnt gene leading to the expression of non-functional mitochondrial nicotinamide nucleotide transhydrogenase, may affect mitochondria-specific features of cellular metabolism, especially NADPH biosynthesis, glutathione reduction and free radical detoxification (1,2) Although the most significant phenotypic variation between C57BL6/J and other BL6 strains is �-cell energy metabolism and insulin release, the effect of Nnt deletion seem to be moderate in nature (3,4). Control C57BL6/J mice lacking functional Nnt gene housed on standard, low-fat diet are still significantly more glucose-tolerant and insulin sensitive from their high-fat diet-fed counterparts (5), which validates the use of C57BL/6J strain in the studies on diet-induced insulin resistance. 

The major factor influencing our choice of C57BL/6J mouse strain over other well established BL6 and DBA/2 strains was both the documented susceptibility to metabolic disorders under HFD feeding and its widely recognized status in insulin resistance and T2D research (6,7). C57BL6/J strain's predisposition to develop insulin resistance makes it a preferred model for studying metabolic disorders induced by high-fat diets (8). Currently C57BL/6J is the most popular BL6 strain used in diabetes-related studies, according to PubMed. Moreover, the significant body of research, which employed modern high-throughput techniques such as genomics, transcriptomics and proteomics was collected using this particular strain, as it was originally used as a reference organism in Mouse Genome Project initiative (9,10). Consequently, the utilization of the C57BL/6J mice, with its established predisposition to glucose intolerance and reduced insulin secretion under high-caloric intake, offers a robust model for investigating metabolic disorders. To address the selection of C57BL6J mice in our study we updated the Introduction section of the manuscript. 

1. Moon D. O. (2023). NADPH Dynamics: Linking Insulin Resistance and β-Cells Ferroptosis in Diabetes Mellitus. International journal of molecular sciences, 25(1), 342. doi.org/10.3390/ijms25010342.

2. Ronchi JA, Figueira TR, Ravagnani FG, Oliveira HC, Vercesi AE, Castilho RF. A spontaneous mutation in the nicotinamide nucleotide transhydrogenase gene of C57BL/6J mice results in mitochondrial redox abnormalities. Free Radic Biol Med. 2013;63:446-456. doi:10.1016/j.freeradbiomed.2013.05.049.

3. Close AF, Chae H, Jonas JC. The lack of functional nicotinamide nucleotide transhydrogenase only moderately contributes to the impairment of glucose tolerance and glucose-stimulated insulin secretion in C57BL/6J vs C57BL/6N mice. Diabetologia. 2021;64(11):2550-2561. doi:10.1007/s00125-021-05548-7.

4. Attané C, Peyot ML, Lussier R, et al. Differential Insulin Secretion of High-Fat Diet-Fed C57BL/6NN and C57BL/6NJ Mice: Implications of Mixed Genetic Background in Metabolic Studies. PLoS One. 2016;11(7):e0159165. Published 2016 Jul 12. doi:10.1371/journal.pone.0159165.

5. Fisher-Wellman KH, Ryan TE, Smith CD, et al. A Direct Comparison of Metabolic Responses to High-Fat Diet in C57BL/6J and C57BL/6NJ Mice. Diabetes. 2016;65(11):3249-3261

6. Montgomery MK, Hallahan NL, Brown SH, et al. Mouse strain-dependent variation in obesity and glucose homeostasis in response to high-fat feeding. Diabetologia. 2013;56(5):1129-1139; 

7. Nguyen-Phuong T, Seo S, Cho BK, Lee JH, Jang J, Park CG. Determination of progressive stages of type 2 diabetes in a 45% high-fat diet-fed C57BL/6J mouse model is achieved by utilizing both fasting blood glucose levels and a 2-hour oral glucose tolerance test. PLoS One. 2023;18(11):e0293888).

8. Collins S, Martin TL, Surwit RS, Robidoux J. Genetic vulnerability to diet-induced obesity in the C57BL/6J mouse: physiological and molecular characteristics. Physiol Behav. 2004;81(2):243-248).

9. Mouse Genome Sequencing Consortium, Waterston RH, Lindblad-Toh K, et al. Initial sequencing and comparative analysis of the mouse genome. Nature. 2002;420(6915):520-562. doi:10.1038/nature01262.

10. Sarsani VK, Raghupathy N, Fiddes IT, et al. The Genome of C57BL/6J "Eve", the Mother of the Laboratory Mouse Genome Reference Strain. G3 (Bethesda). 2019;9(6):1795-1805. Published 2019 Jun 5. doi:10.1534/g3.119.400071.

2. The authors do not specify whether CPT2 is altered under their experimental conditions or if a compensatory mechanism occurs in HFD(+Acsl1) and HFD(-Acsl1).

Author’s response: Thank You, Reviewer, for Your insightful comment. Based on the literature, we selected CPT1b (membrane carnitine palmitoyltransferase 1, muscular isoform) as a marker enzyme for mitochondrial fatty acids import. This isoform is widely recognized as a crucial rate-limiting enzyme that regulates the rate of long-chain fatty acid entry into the mitochondrial β-oxidation (1). Therefore, we opted to focus on presenting CPT1b protein expression in our analysis, considering its significance to our research objectives. Consequently, we determined that performing additional CPT2 protein analysis would be excessive, due to the limited amount of remaining material available for experimentation.

1. Bruce CR, Hoy AJ, Turner N, et al. Overexpression of carnitine palmitoyltransferase-1 in skeletal muscle is sufficient to enhance fatty acid oxidation and improve high-fat diet-induced insulin resistance. Diabetes. 2009;58(3):550-558).

3. In Figure 4, the authors measure the phosphorylation of Akt/PKB by assessing the phosphorylation of Akt Ser473 using western blotting. It would be valuable to know if they also conducted western blot analysis with an antibody targeting Thr308-Akt since pAKT/PKB requires activation on both sites (Thr308 and Ser473) for full signaling pathway activity.

Author’s response: We appreciate Your valuable comment. Following the Reviewer's suggestion, we analyzed Akt Thr308 phosphorylation. The results presented in Figure 4 show, that Acsl1 partial ablation is connected with AKT phosphorylation at both the Thr308 and Ser473 sites. This finding further confirms the impact of muscle Acls1 downregulation on the activity of insulin signaling pathway.

Minor comments:

1. The fig

---

## [Decision Letter · Decision Letter 2]

12 Jul 2024

shRNA-mediated down-regulation of Acsl1 reverses skeletal muscle insulin resistance in obese C57BL6/J mice

PONE-D-23-38615R2

Dear Dr. Piotr Zabielski,,

We’re pleased to inform you that your manuscript has been judged scientifically suitable for publication and will be formally accepted for publication once it meets all outstanding technical requirements.

Kind regards,

Herve Le Stunff

Academic Editor

PLOS ONE

Additional Editor Comments (optional):

Reviewers' comments:

Reviewer's Responses to Questions

**Comments to the Author**

1. If the authors have adequately addressed your comments raised in a previous round of review and you feel that this manuscript is now acceptable for publication, you may indicate that here to bypass the “Comments to the Author” section, enter your conflict of interest statement in the “Confidential to Editor” section, and submit your "Accept" recommendation.

Reviewer #3: All comments have been addressed

Reviewer #4: All comments have been addressed

2. Is the manuscript technically sound, and do the data support the conclusions?

Reviewer #3: Yes

Reviewer #4: Yes

3. Has the statistical analysis been performed appropriately and rigorously? 

Reviewer #3: Yes

Reviewer #4: Yes

4. Have the authors made all data underlying the findings in their manuscript fully available?

Reviewer #3: Yes

Reviewer #4: Yes

5. Is the manuscript presented in an intelligible fashion and written in standard English?

Reviewer #3: Yes

Reviewer #4: Yes

6. Review Comments to the Author

Reviewer #3: The authors responded to my questions/comments in the best possible way. I now believe that the study is strong enough for publication in PloS One.

Reviewer #4: In this new submission of Roszczyc-Owsiejczuk et al., the authors have taken into account the comments of the various reviewers, enriched the article introduction, added important data by performing new PCRs and western blots and discussed their statistical analyses.

These improvements make the article easier to read and reinforce the results and conclusions demonstrated by the authors.

7. PLOS authors have the option to publish the peer review history of their article (what does this mean?). If published, this will include your full peer review and any attached files.

Reviewer #3: No

Reviewer #4: No

---

## [Editor Report · Acceptance letter]

16 Jul 2024

PONE-D-23-38615R2 

PLOS ONE

Dear Dr. Zabielski, 

I'm pleased to inform you that your manuscript has been deemed suitable for publication in PLOS ONE. Congratulations! Your manuscript is now being handed over to our production team.

Kind regards, 

on behalf of

Dr. Herve Le Stunff 

Academic Editor

PLOS ONE